# Charge-loop current order and $Z_3$ nematicity mediated by bond order fluctuations in kagome metals

Rina Tazai [1] ✉, Youichi Yamakawa [2] & Hiroshi Kontani [2]

Recent experiments on geometrically frustrated kagome metal $AV_3Sb_5$ ($A$ = K, Rb, Cs) have revealed the emergence of the charge loop current (cLC) order near the bond order (BO) phase. However, the origin of the cLC and its interplay with other phases have been uncovered. Here, we propose a novel mechanism of the cLC state, by focusing on the BO phase common in kagome metals. The BO fluctuations in kagome metals, which emerges due to the Coulomb interaction and the electron-phonon coupling, mediate the odd-parity particle-hole condensation that gives rise to the topological current order. Furthermore, the predicted cLC+BO phase gives rise to the $Z_3$-nematic state in addition to the giant anomalous Hall effect. The present theory predicts the close relationship between the cLC, the BO, and the nematicity, which is significant to understand the cascade of quantum electron states in kagome metals. The present scenario provides a natural understanding.

Recent discovery of the kagome-lattice metal $AV_3Sb_5$ ($A$ = K, Rb, Cs) shown in Fig. 1a has opened the way to study the unique physics of geometrically frustrated metals with strong correlation[1–3]. In $CsV_3Sb_5$, the formation of the 2 × 2 Star-of-David or Tri-Hexagonal density wave (DW) was detected by scanning tunneling microscopy (STM) at $T \approx 90$ K at ambient pressure[4,5]. It is presumably the triple-**q** ($3Q$) bond order (BO) shown in Fig. 1b, which is the even-parity modulation in the hopping integral $\delta t_{ij}^{b}$ (=real)[6–12]. Below the BO transition temperature $T_{BO}$, superconductivity (SC) with highly anisotropic gap emerges for $A$ = Cs[13,14], and the gap structure changes to isotropic by introducing impurities. Also, nodal to nodeless crossover is induced by the external pressure in $A$ = Rb,K[15]. These results are naturally understood based on the BO fluctuation mechanism[12].

More recently, the non-trivial time reversal symmetry breaking (TRSB) order at $T_{TRSB}$ attracts considerable attention. It has been reported by $\mu$SR study[15–18], Kerr rotation analysis[19], field-tuned chiral transport study[20], and STM measurements[4,20]. The transition temperature $T_{TRSB}$ is close to $T_{BO}$ in many experiments, while the TRSB order parameter is strongly magnified at $T^* \approx 35$ K for $A$ = Cs[16,18,20] and $T^* \approx 50$ K for $A$ = Rb[15]. Recently, magnetic torque measurement reveals

the TRSB order associated with the rotational symmetry breaking, which is called the nematic order, at $T^* \approx 130$ K[21]. In contrast, TRSB was not reported by different experimental groups using the Kerr rotation[22] and STM[23] measurements. Thus, the TRSB onset temperature is still under debate. The chiral cLC is driven by the additional odd-parity hopping integral $\delta t_{ij}^{c}$ (=imaginary), and the accompanied topological charge-current[24] gives the giant anomalous Hall effect (AHE) below $T \approx 35$ K[25,26]. The correlation-driven topological phase in kagome metals is very unique, while its mechanism is still unknown.

In addition to the cascade of quantum phase transitions, the emergent nematic order inside the BO and the cLC phases attracts great attention. The nematic transition is clearly observed by the elastoresistance[27], the scanning birefringence[19], and the STM[5] studies. In addition, nematic SC states have been reported[23,28]. Thus, kagome metals provide a promising platform for exploring the interplay between electron correlations and topological nature.

To understand the rich quantum phases in kagome metals, lots of theoretical studies have been performed[6–12,29–31]. Each BO and cLC order is explained by introducing various off-site interactions in the mean-field approximation (MFA)[9,29,32,33], while a fine-tuning of off-site

[1]Yukawa Institute for Theoretical Physics, Kyoto University, Kyoto 606-8502, Japan. [2]Department of Physics, Nagoya University, Furo-cho, Nagoya 464-8602, Japan. ✉e-mail: rina.tazai@yukawa.kyoto-u.ac.jp

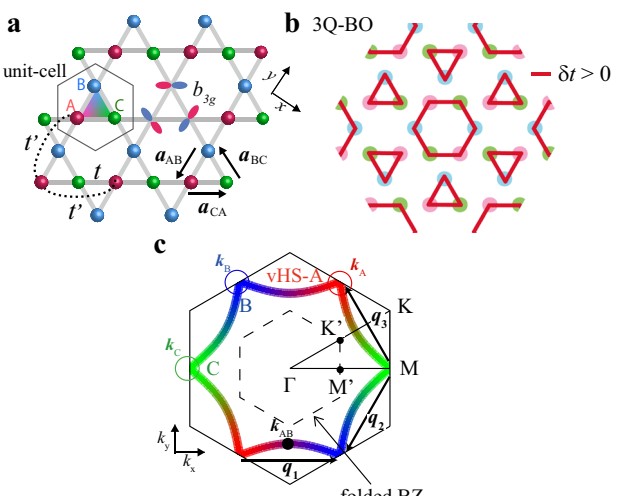

**Fig. 1 | Lattice structure, Fermi surface, and BO form factor in kagome metal.**
**a** Kagome-lattice structure composed of the sublattices A, B, and C. $2\mathbf{a}_{lm}$ is the minimum translation vector, and we set $|2\mathbf{a}_{lm}| = 1$. The relation $\mathbf{a}_{AB} + \mathbf{a}_{BC} + \mathbf{a}_{CA} = \mathbf{0}$ holds. **b** $3Q$ Tri-Hexagonal bond order (BO) state. **c** Fermi surface (FS) at $n = 0.917$ with the nesting vectors $\mathbf{q}_1$, $\mathbf{q}_2$, $\mathbf{q}_3$. The color of the FS represents the weight of the sublattice (A = red, B = blue, C = green). The FS has single sublattice character near the van-Hove singularity (vHS) points. In kagome metals, $\mathbf{q}_1$ connects vHS-A and vHS-B. It is given as $\mathbf{q}_1 = (2\mathbf{a}_{AB}) \times (2\pi/\sqrt{3})\mathbf{e}_z$, where $\mathbf{e}_z$ is the unit vector perpendicular to the $xy$-plane.

interactions is necessary to explain the cascade of phase transitions. On the other hand, beyond-MFA mechanisms have been developed to explain the rich phase transitions[34–45]. For example, strong interplay between the magnetism, nematicity and SC in Fe-based superconductors and other strongly correlated metals were understood by beyond-MFA mechanisms[36–45]. It is urgent and important to elucidate why the BO and cLC orders/fluctuations coexist in the study of kagome metals. For example, these fluctuations will mediate non-BCS SC[12] and exotic pair-density-wave states[46–48].

In this paper, we reveal that the cLC order is mediated by the BO fluctuations that are abundant above $T_{BO}$ in kagome metals[49,50]. The sizable off-site Umklapp scattering by the BO fluctuations induces the odd-parity and TRSB current order (=imaginary $\delta t_{ij}^c$). This cLC mechanism is universal because it is irrelevant to the origin of the BO. Furthermore, we discover that the coexistence of the BO and the cLC order gives rise to the novel $Z_3$ nematicity along the three lattice directions reported in refs. 5,19,27. The present theory reveals the close relationship between the cLC, BO, nematicity, and SC state, which is significant to understand the unsolved quantum phase transitions in kagome metals.

The phase transitions in metals are described as the symmetry breaking of the normal self-energy; $\Delta\Sigma \equiv \Sigma - \Sigma_{A_{1g}}$[45,51]. $\Delta\Sigma$ is determined by the stationary condition of the free energy; $\delta F[\Delta\Sigma]/\delta(\Delta\Sigma) = 0$. The DW equation enables us to derive the solution that satisfies the stationary condition, as we proved based on the Luttinger-Ward theory[51]. Based on the DW equation, we discover that the odd-parity and TRSB $\Delta\Sigma$ is driven by the BO fluctuation exchange processes. (Note that the DW equation for $\Delta\Sigma$ is analogous to the Eliashberg equation for the SC gap $\Delta$).

## Results

### BO form factor and fluctuations

Here, we introduce the kagome-lattice tight-binding model with a single $d$-orbital of each vanadium site (A, B, or C) shown in Fig. 1a. (The $d$-orbital belongs to $b_{3g}$ of the $D_{2h}$ point group at V site, while its representation is not essential here). The kinetic term is given by

$\hat{H}_0 = \sum_{\mathbf{k},l,m,\sigma} h_{lm}^0(\mathbf{k}) c_{\mathbf{k},l,\sigma}^{\dagger} c_{\mathbf{k},m,\sigma}$, where $l$, $m$ denote the sublattices A, B, C, and $h_{lm}^0(\mathbf{k})$ ($= h_{ml}^0(\mathbf{k})^*$) is the Fourier transform of the nearest-neighbor hopping integral $t$ in ref. 52 in addition to the inter-sublattice hopping $t'$ shown in Fig. 1a. We set $t$ ($=-0.5$eV) to fit the bandwidth, and $t'$ ($= -0.08$ eV) to reproduce the shape of the Fermi surface (FS). Numerical results are insensitive to the presence of $t'$. Hereafter, the unit of energy is eV unless otherwise noted. The FS around the van-Hove singularity (vHS) point ($\mathbf{k} \approx \mathbf{k}_A$, $\mathbf{k}_B$, or $\mathbf{k}_C$) is composed of a single $3d$-orbital on V ion, which is called the sublattice interference[6]. This simple three-site model well captures the main pure-type FS in kagome metals[4,53–57]. The FS at the vHS filling ($n_{vHS} = 0.917$ per site and both spins) is shown in Fig. 1c. The wavevectors of the BO correspond to the inter-sublattice nesting vectors $\mathbf{q}_n$ ($n = 1, 2, 3$) in Fig. 1c. (The equivalent square lattice kagome model is convenient for the numerical study; see Supplementary Note 1). The good inter-sublattice nesting of the FS naturally triggers the observed inter-sublattice BO at $\mathbf{q} = \mathbf{q}_n$, as shown in previous theoretical studies[6,7,12].

The Fourier transform of the BO modulation, $\delta t_{ij}^b$, gives the even-parity BO form factor $g_{\mathbf{q}}^{lm}(\mathbf{k})$[45,58]:

$$g_{\mathbf{q}}^{lm}(\mathbf{k}) = \frac{1}{N} \sum_{i}^{sub-l} \sum_{j}^{sub-m} \delta t_{ij}^b e^{i\mathbf{k}\cdot(\mathbf{r}_i - \mathbf{r}_j)} e^{-i\mathbf{q}\cdot\mathbf{r}_j}, \qquad (1)$$

where $\mathbf{q}$ is the wavevector of the BO. In this study, we use the simplified BO form factor due to the nearest sites presented in Supplementary Note 2-1. The form factor at $\mathbf{q} = \mathbf{q}_1$, $g_{\mathbf{q}_1}^{lm}$, is nonzero only when $\{l, m\} = \{A, B\}$, and we set $g_{\mathbf{q}}^{lm} = g_{\mathbf{q}_1}^{lm}$ when $\mathbf{q}$ is in region I in Fig. 2b. In the same way, we set $g_{\mathbf{q}}^{lm} = g_{\mathbf{q}_2}^{lm}$ ($g_{\mathbf{q}_3}^{lm}$) when $\mathbf{q}$ is in region II (III). $g_{\mathbf{q}_2}^{lm}$ ($g_{\mathbf{q}_3}^{lm}$) is nonzero for $\{l, m\} = \{B, C\}$ ($\{C, A\}$). This treatment is justified because the BO fluctuations strongly develop only for $\mathbf{q} \approx \mathbf{q}_n$ in kagome metals. Furthermore, we use $\bar{f}_{\mathbf{q}}(\mathbf{k}) = (f_{\mathbf{q}_n}(\mathbf{k}) + f_{\mathbf{q}_n}(\mathbf{k} + \mathbf{q} - \mathbf{q}_n))/2$ for $\mathbf{q} \sim \mathbf{q}_n$ in the numerical study to improve the accuracy. Both BO and cLC form factors are Hermite $\delta t_{ij}^{lm} = (\delta t_{ji}^{ml})^*$, which leads to the relation $g_{\mathbf{q}}^{lm}(\mathbf{k}) = (g_{-\mathbf{q}}^{ml}(\mathbf{k} + \mathbf{q}))^*$[45].

To express the development of the bond order and fluctuations in kagome metals, we introduce the following effective BO interaction:

$$\hat{H}_{int} = -\frac{1}{2N} \sum_{\mathbf{q}} \frac{v}{2} \hat{O}_{\mathbf{q}}^g \hat{O}_{-\mathbf{q}}^g, \qquad (2)$$

where $\hat{O}_{\mathbf{q}}^g \equiv \sum_{\mathbf{k},l,m,\sigma} g_{\mathbf{q}}^{lm}(\mathbf{k}) c_{\mathbf{k}+\mathbf{q},l,\sigma}^{\dagger} c_{\mathbf{k},m,\sigma}$ is the BO operator[45,58,59]. and $v$ is the effective interaction. We assume that the form factor $g_{\mathbf{q}}^{lm}(\mathbf{k})$ is normalized as $\max_{\mathbf{k},l,m} |g_{\mathbf{q}}^{lm}(\mathbf{k})| = 1$ at each $\mathbf{q}$, i.e., $|\delta t_{ij}^b| \equiv 1/2$ for the nearest sites. Then, the maximum matrix element of BO interaction in Eq. (2) is $v/2$. The interaction (2) would originate from the combination of (i) the paramagnon-interference due to on-site $U$[12], (ii) the bond-stretching phonon[60], and (iii) the Fock term of off-site Coulomb interaction $V^{\rho}$. In (i), Eq. (2) is induced by the spin-fluctuation-mediated beyond-RPA processes, whose diagrammatic expressions are shown in Fig. 3c, d in ref. 12. This processes give rise to the nematic BO in Fe-based SCs[45]. A great advantage of this theory[12] is that the function of the BO form factor and the BO wavevector are automatically optimized to maximize $T_{BO}$. Based on this theory, the BO at $\mathbf{q} = \mathbf{q}_n$ ($n = 1, 2, 3$) is robustly obtained based on the first principles multi-orbital model for $CsV_3Sb_5$[12]. The effective parameter $v$ in Eq. (2) is given as $v_{AL} \sim [g_{back} + g_{um}]/2$, which is about 1.5 near the BO critical point ($\lambda_{bond} \lesssim 1$), as we see in Fig. 3e of ref. 12. Thus, the value of $v$ given by the AL processes is comparable to that used in the present study. In (ii), $g_{\mathbf{q}}^{lm}(\mathbf{k})$ is given by the hopping modulation due to the stretching mode and $v = 2\eta^2/\omega_D$, where $\eta$ is the electron-phonon ($e$-ph) coupling constant and $\omega_D$ is the phonon energy at $\mathbf{q} \approx \mathbf{q}_n$. The BO interaction

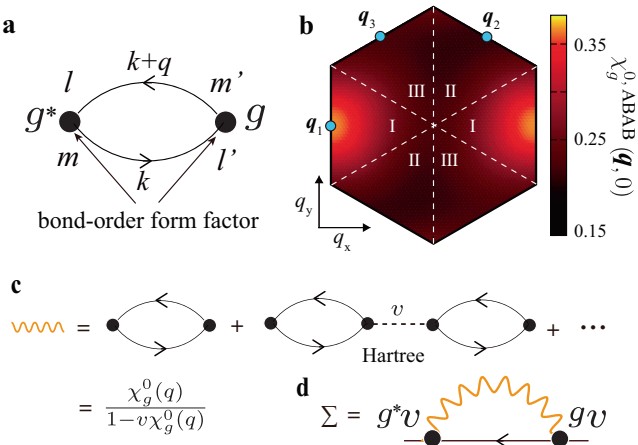

**Fig. 2 | BO fluctuations and self-energy. a** Expressions of the bond-order (BO) irreducible susceptibility $\chi_g^{0,lmm'l'}(q)$. **b** Obtained **q**-dependence for $\chi_g^{0,\mathrm{ABAB}}(q)$, which takes the maximum at $\mathbf{q} = \mathbf{q}_1$. **c** $\hat{\chi}_g(q)$ enlarged by the Hartree term of the electron-phonon interaction (2). [Note that $\hat{\chi}_g(q)$ is also enlarged by the Fock term of the off-site Coulomb interaction; see Supplementary Note 2-2]. **d** Self-energy induced by the BO fluctuations.

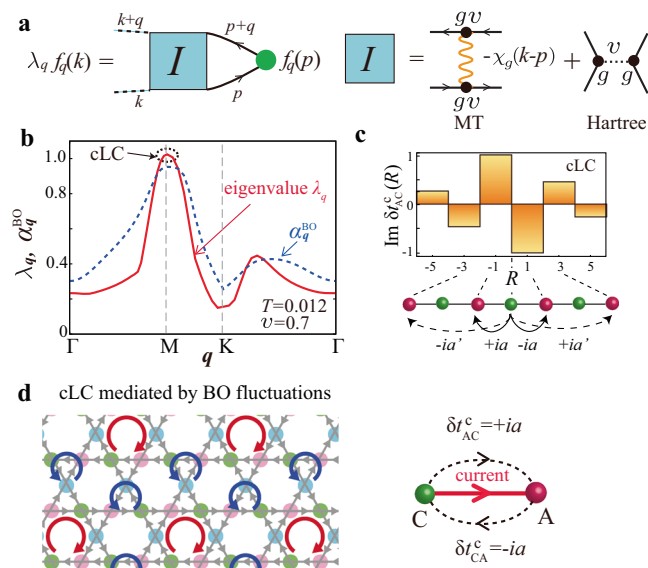

**Fig. 3 | cLC order driven by BO fluctuation mechanism. a** Density-wave (DW) equation due to the single exchange term of the bond-order (BO) fluctuations. **b** Eigenvalue of the DW equation $\lambda_\mathbf{q}$ (red-solid line) and BO Stoner factor $\alpha_\mathbf{q}^{\mathrm{BO}}$ (blue-dashed line) for $v = 0.7$ and $T = 0.012$. Both show peaks at $\mathbf{q} = \mathbf{q}_n$. **c** Imaginary hopping modulation $\mathrm{Im}\,\delta t_{\mathrm{AC}}^c(R)$. Its triple-**q** order gives the cLC pattern in (**d**). One can check that the clock-wise (anti-clock-wise) loop currents on hexagons (triangles) in (**d**) are inverted and moved by $\mathbf{a}_{\mathrm{AC}}$ under the sign change of $f_{\mathbf{q}_3}^{\mathrm{AC}}$.

for the three vHS points model was derived in ref. 10. In (iii), $v = 2V$ as we explain in the Supplementary Note 2-2. Thus, the effective interaction (2) is general. A possible driving forces of the BO have been discussed experimentally[61,62].

Next, we study the susceptibility of the BO operator (per spin) defined as

$$\chi_g(\mathbf{q}, \omega_l) \equiv \frac{1}{2} \int_0^\beta d\tau \left\langle \hat{O}_\mathbf{q}^g(\tau) \hat{O}_{-\mathbf{q}}^g(0) \right\rangle e^{i\omega_l \tau} \quad (3)$$

where $\omega_l$ is a boson Matsubara frequency. $\hat{O}_\mathbf{q}^g(\tau)$ is the Heisenberg representation of the BO operator. When $v = 0$, $\chi_g(q)$ is equivalent to the BO irreducible susceptibility $\chi_g^0(q)$[45,58]:

$$\chi_g^0(q) = \sum_{lmm'l'} \chi_g^{0,lmm'l'}(q), \quad (4)$$

$$\chi_g^{0,lmm'l'}(\mathbf{q}, \omega_l) = \frac{T}{N} \sum_{\mathbf{k}, \epsilon_n} g_\mathbf{q}^{lm}(\mathbf{k})^* G_{lm'}(\mathbf{k} + \mathbf{q}, \epsilon_n + \omega_l) \\ \times G_{l'm}(\mathbf{k}, \epsilon_n) g_\mathbf{q}^{m'l'}(\mathbf{k}), \quad (5)$$

where $q \equiv (\mathbf{q}, \omega_l = 2\pi Tl)$ and $\epsilon_n$ is a fermion Matsubara frequency. Equation (5) contains two form factors, so it vanishes when $l = m$ or $l' = m'$. Its diagrammatic expression is given in Fig. 2a. The numerical result for $\chi_g^{0,\mathrm{ABAB}}(\mathbf{q}, 0)$ is shown in Fig. 2b, which exhibits the broad peak at the nesting vector between vHS-A and vHS-B; $\mathbf{q} = \mathbf{q}_1$.

The BO susceptibility in Eq. (3) is strongly magnified by the Hartree term of Eq. (2) because of the same form factors in both equations. Its process is expressed in Fig. 2c, and its analytic expression is

$$\chi_g(q) = \chi_g^0(q)/(1 - v\chi_g^0(q)). \quad (6)$$

where the notation $q \equiv (\mathbf{q}, \omega_l = 2\pi Tl)$ is used. Here, the relation $\chi_g(\mathbf{q}_n, 0) \propto (1 - \alpha_{\mathrm{BO}})^{-1}$ holds, where $\alpha_{\mathrm{BO}} \equiv \max_\mathbf{q} v\chi_g^0(\mathbf{q})$ is the BO stoner factor. $\chi_g(\mathbf{q}_n, 0)$ diverges when $\alpha_{\mathrm{BO}} = 1$. In contrast, the cLC susceptibility for the odd-parity cLC form factor, $f_\mathbf{q}^{lm}(\mathbf{k} - \mathbf{q}/2) = -f_\mathbf{q}^{ml}(-\mathbf{k} - \mathbf{q}/2)$, is unchanged by the Hartree term because $g$ is even-parity.

The BO susceptibility is the largest in the Hartree-Fock (HF) approximation. As we discuss in the Supplementary Note 3, the BO and

cLC susceptibilities at $\mathbf{q} = \mathbf{q}_n$ are $\tilde{\chi}_g \propto (1 - (v+v')\chi_g^0)^{-1}$ and $\tilde{\chi}_{\mathrm{cLC}} \propto (1 - v''\chi_g^0)^{-1}$, respectively. Here, $-v' \sim v'' \sim 0.3yv$ originates from the Fock term. (The coefficient $y (\sim O(1))$ depends on the origin of BO fluctuations. $y = 1/2$ for $H_{\mathrm{imt}}$ in Eq. (2). The detailed discussion on $y$ will be presented later). Thus, both susceptibilities are enlarged, while $\tilde{\chi}_{\mathrm{cLC}} < \tilde{\chi}_g$ within the HF approximation. However, we discover that $-v'$ and $v''$ are further enlarged by the Maki-Thompson (MT) vertex corrections.

The MT term describes the scattering of electrons due to the developed bosonic fluctuations. This scattering process is important in metals near the quantum critical points. For example, in nearly anti-ferromagnetic metals, the $d$-wave SC transition is induced by the MT processes of spin fluctuations. In kagome metals, the MT term represents the strong inter vHS scattering of electrons mediated by the abundant BO fluctuations; see Fig. 3a. (The MT term also describes the $s$-wave SC state in kagome metals[12]). Here, we find that both $\tilde{\chi}_g$ and $\tilde{\chi}_{\mathrm{cLC}}$ are comparably enlarged due to the MT processes in the present theory.

To understand the BO+cLC phase diagram and the energy scale of these orders accurately, we have to include the self-energy that describes the quasiparticle properties. We calculate the on-site self-energy due to BO fluctuations (see Eq. (8) in "Methods"). The fluctuation-induced self-energy is essential to reproduce the $T$-dependence of various physical quantities, as well-known in spin fluctuation theories[63-65]. Here, we calculate $\chi_g(q)$ in Eq. (6) and $\Sigma_m(\epsilon_n)$ in Eq. (8) self-consistently.

## BO fluctuation-mediated cLC order

Next, we discuss the cLC mechanism. The HF approximation for the BO interaction (2) does not lead to the cLC order, as we explain in the Supplementary Note 3. (It is the same for off-site Coulomb interaction case; see Supplementary Note 2-1). Thus, the cLC order should be ascribed to the beyond-HF mechanism. Here, we explain that the strong electron scattering between different vHS points due to the BO fluctuations, which are described as the MT processes, causes the odd-parity cLC order $\delta t_{ij}^c = -\delta t_{ji}^c$. (Note that the spin-fluctuation-exchange

processes cause the cLC order in quasi-1D systems[66]). This process is generated by solving the following linearized DW equation[41,51,66]:

$$\lambda_{\mathbf{q}} f_{\mathbf{q}}^L(k) = \frac{T}{N} \sum_{p, M_1, M_2} I_{\mathbf{q}}^{L, M_1}(k, p) \tag{7}$$
$$\times \{-G(p)G(p+\mathbf{q})\}^{M_1, M_2} f_{\mathbf{q}}^{M_2}(p),$$

where $L \equiv (l, l')$ and $M_i$ represent the pair of sublattice indices. $I_{\mathbf{q}}^{L, M}(k, p) \propto -\chi_g(k - p)$ is given by the BO fluctuation scattering process shown in Fig. 3a, which is called the MT process. The expression of $I_{\mathbf{q}}^{L, M}$ is given in Eq. (10) in "Methods" section. Note that $T\sum_n \{-G(\mathbf{p}, \epsilon_n) G(\mathbf{p} + \mathbf{q}, \epsilon_n)\} > 0$.

By solving the DW equation (7), the optimized order parameter function is given as the eigenfunction $f_{\mathbf{q}}^L(k)$ for the maximum eigenvalue $\lambda_{\mathbf{q}}$. $\max\{\lambda_{\mathbf{q}}\} = 1$ at the phase transition temperature. Note that $f_{\mathbf{q}}^L(k)$ represents the symmetry-breaking part in the normal self-energy $\Delta\Sigma(\mathbf{k}, \mathbf{q}) \sim \langle c_{\mathbf{k}+qo}^\dagger c_{\mathbf{k}\sigma} \rangle$, and DW equation is directly derived from the stationary condition $\delta F[\Delta\Sigma]/\delta(\Delta\Sigma) = 0$[51]. We can regard the DW equation (7) as the gap equation for the optimized particle-hole (p-h) condensation[45,51].

Note that the BO fluctuation-mediated interaction for the self-energy (Eq. (8)) and that for the kernel function (Eq. (10)) have the same coefficient $y$, guaranteed by the Ward identity. The $y$ depends on the BO fluctuation mechanism: $y = 1/2$ for the BO interaction $v$ in Eq. (2) that works only in charge-channel. $y \approx 1/2$ for the AL mechanism for the same reason[12]. $y = 2$ for the off-site $V$ that induced both charge- and three spin-channel BO fluctuations as we explain in Supplementary Note 2-2. In kagome metals, both $v$ and $V$ coexist. In this case, BO fluctuations in charge-channel dominate over those in spin-channel, and therefore $y \gtrsim 0.5$ is expected in real kagome metals. Detailed explanation is given in Supplementary Note 2-3. Because we are interested in a general argument, we set $y = 0.5 \sim 1$ as a model parameter below. Note that the Aslaoazov-Larkin term is unimportant as we discuss in Supplementary Note 4.

Figure 3b shows the largest eigenvalue of the DW equation $\lambda_{\mathbf{q}}$ (red line) and BO Stoner factor $a_{\mathbf{q}}^{BO} \equiv v\chi_g^0(\mathbf{q})$ (blue line) as functions of $\mathbf{q}$, for $v = 0.7$ and $T = 0.012$. They exhibit the maximum value at $\mathbf{q} = \mathbf{q}_n$ ($n = 1, 2, 3$). The corresponding solution of the DW equation is odd-parity: $f_{\mathbf{q}}^{lm}(\mathbf{k} - \mathbf{q}/2) = -f_{\mathbf{q}}^{ml}(-\mathbf{k} - \mathbf{q}/2)$. Then, the corresponding real-space hopping modulation is odd-parity $\delta t_{ij}^c = -\delta t_{ji}^c$ and pure imaginary when $\delta t_{ij}^c$ is Hermitian. The obtained $\delta t_{AC}^c(R) \equiv \delta t_{i_A j_C}^c$ for the cLC at $\mathbf{q} = \mathbf{q}_3$ along the A-C direction is shown in Fig. 3c, where the odd integer $R$ is defined as $\mathbf{r}_i^C - \mathbf{r}_j^A \equiv R\mathbf{a}$. In addition, the odd-parity relation $\delta t_{AC}^c(R) = -\delta t_{CA}^c(-R)$ is verified. The obtained charge loop current pattern for the 3Q state is depicted in Fig. 3d.

Here, we discuss why the cLC order is mediated by the BO fluctuations. Let us consider the infinite series of MT terms in Fig. 4a, which is equal to $f_{\mathbf{q}_3}^{AC}(\lambda_{\mathbf{q}_3}^{-1} - 1)^{-1}$ according to the DW equation (7). The first term together with other odd-order MT terms in Fig. 4a give the repulsive Umklapp interaction, $\Gamma_{um}^{MT} < 0$, which leads to the odd-parity order $f_{\mathbf{q}_3}^{AC} = -f_{\mathbf{q}_3}^{CA}$. In contrast, the second term together with other even-order MT terms give the attractive backward interaction, $\Gamma_{back}^{MT} > 0$, which gives the attraction among the same $f_{\mathbf{q}_3}^{lm}$. Therefore, all series of MT terms cooperatively induce the odd-parity current order form factor shown in Fig. 3c. Figure 4b exhibits the obtained $v$-dependence of the cLC eigenvalue as a function of $T$ in the case of $y = 1$.

Figure 4c exhibits the $T$-dependence of $\lambda_{\mathbf{q}_3}$ for $v = 0.4$–1.4 at $y = 1$. The cLC transition temperature $T_{cLC}$ is given by the relation $\lambda_{\mathbf{q}_3} = 1$. The color on each line represents $\alpha_{BO}$: It is clearly seen that $\alpha_{BO}$ at $T = T_{cLC}$ monotonically increases with $v$. In 2D systems, $\alpha_{BO}$ asymptotically approaches 1 with $v$, but never exceeds 1 due to the $\chi_g$-induced self-energy[65,67,68]. Here, $T_{BO}$ is defined as $\alpha_{BO} = \alpha_{BO}^*$ with $\alpha_{BO}^* = 0.985$, which is shown as a small circle on each line in Fig. 4c, by considering the

small inter-layer BO coupling $|v_\perp|(\ll v)$. (Overall results are unchanged for $\alpha_{BO}^* \sim 0.99$). The three-dimensional (3D) BO appears when $\chi_g^{3D} = \chi_g^{2D}/(1 - |v_\perp| \chi_g^{2D}) = \infty$, that is, $|v_\perp| \sim (1 - \alpha_{BO})v$. Similar method is frequently used in deriving $T_{SDW}$ in spin fluctuation theories[67,68]. When $v$ is small, the relation $T_{cLC} > T_{BO}$ holds, which is natural because the MT term becomes large for $\alpha_{BO} \lesssim 1$. With increasing $v$, however, the opposite relation $T_{cLC} < T_{BO}$ is realized due to the large self-energy effect.

The obtained $T_{cLC}$ and $T_{BO}$ as functions of $v$ are shown in Fig. 4d $y = 1$ and e $y = 0.5$. In d, $T_{cLC} = T_{BO}$ is realized at $v = v^* \approx 1.03$, and $T_{cLC}/T_{BO} > 1$ is realized in the weak-coupling region $v < v^*$. The opposite relation $T_{cLC}/T_{BO} < 1$ is obtained in the strong-coupling region $v > v^*$ because the eigenvalue of DW equation (7) is suppressed by the large self-energy. In e, $T_{cLC} = T_{BO}$ at $v = v^* \approx 0.55$.

Figure 4d, e indicate that both BO and cLC instabilities are comparable for $v^* \approx v$. Based on the parity argument, the BO (cLC) instability is given by $\Gamma_{back} + (-)\Gamma_{um}$. Therefore, the relation $\Gamma_{back} \gg |\Gamma_{um}|$ should be satisfied for $v^* \approx v$. In fact, the Hartree process gives positive $\Gamma_{back}^H = \Gamma_{um}^H \sim v/(1 - \chi_g^0)$, so the Hartree and MT processes strengthen each other in $\Gamma_{back}$ but cancel each other in $\Gamma_{um}$. This relation is verified by the parquet RG study in Supplementary Note 5.

We discuss that Fig. 4d, e naturally explain the experimental $P$-$T$ phase diagram with $T_{BO}$ and $T_2^* (\sim T_{TRSB})$ given by $\mu$SR study[15] for $A = Rb$, considering that $v/W_{band}$ decreases with $P$. A schematic BO+cLC phase diagram derived from the present theory is depicted in Fig. 4f. (This schematic phase diagram is supported by the Ginzburg-Landau (GL) analysis in Supplemental Fig. 11a–c. The suppression of the secondary order due to the primary order is considered). The cLC phase is realized next to the BO phase because it is mediated by the BO fluctuations. This cLC+BO phase diagram is reminiscent of the SC-SDW phase diagram of spin-fluctuation-mediated superconductors, which has been reproduced by considering the self-energy[67,68].

### $Z_3$-nematic state given by the cLC-BO coexistence

To understand the cLC+BO coexisting states in Fig. 4f, the Ginzburg-Landau (GL) free energy analysis is very useful[10,30,31]. For example, the third-order GL term is $F^{(3)} = b_1\phi_1\phi_2\phi_3 + b_2(\phi_1\eta_2\eta_3 + \eta_1\phi_2\eta_3 + \eta_1\eta_2\phi_3)$, where the coefficients satisfy the relation $b_1 \sim -b_2$, and $(\phi_1, \phi_2, \phi_3)$ [$(\eta_1, \eta_2, \eta_3)$] is the magnitude of the BO [cLC] parameter at $\mathbf{q} = \mathbf{q}_1, \mathbf{q}_2, \mathbf{q}_3$. Here, we introduce the 3Q states $\boldsymbol{\phi}_1 = (\phi/\sqrt{3})(1,1,1)$, $\boldsymbol{\eta}_1 = (\eta/\sqrt{3})(1,1,1)$, and $\boldsymbol{\eta}_2 = (\eta/\sqrt{3})(1, -1, -1)$. The chiral center of $\boldsymbol{\eta}_1$ coincides with the center of the BO $\boldsymbol{\phi}_1$, while the center of $\boldsymbol{\eta}_2$ is shifted by $\mathbf{a}_{BA}$ from that of $\phi_1$. Thus, the coexisting state $(\boldsymbol{\phi}_1, \boldsymbol{\eta}_{1[2]})$ has the $C_{6[2]}$-symmetry as shown in Fig. 5a [c], and its FS in the folded Brillouin zone is in Fig. 5b [d]. As we explain in Supplementary Note 6-1, $F^{(3)}$ for the $C_2$-coexisting state is lower than that for the $C_6$-coexisting state in the case of $|\phi| \gg |\eta|$ for fixed $|\phi|, |\eta|$. (The optimized cLC order in the $C_2$-phase is $\boldsymbol{\eta}_2' \propto (2, -1, -1)$; see Supplementary Note 6-1). Therefore, the BO+cLC $Z_3$ nematic state is realized when $v > v^*$. This result is consistent with the recent observation of out-of-phase combination of bond charge order and loop currents by STM measurement[69]. We comment that the nematic BO+cLC phase is obtained when $T_{BO} \gtrsim T_{cLC}$ by minimizing the GL free energy $F[\boldsymbol{\phi}, \boldsymbol{\eta}]$ exactly in Supplementary Note 6-2.

We also discuss the case of $v < v^*$, where cLC is the primary order as shown in Fig. 4f. The $C_6$ symmetry 3Q cLC order appears at $T = T_{cLC}$ when $2d_{2,a}/d_{2,b} > 1$ as we discuss in Supplementary Note 5, where $d_{2,a}$ ($d_{2,b}$) is the GL coefficients of the $\eta_1^4$ ($\eta_1^2\eta_2^2$) term. Note that the primary 3Q cLC order induces the secondary BO parameter even above $T_{BO}$ through the $b_2$-term in $F^{(3)}$ [10]. In contrast, the 1Q cLC state is realized when $2d_{2,a}/d_{2,b}$ is smaller than unity. Thus, the electronic state becomes nematic at $T_{cLC}(>T_{BO})$. In this case, there is no secondary BO component above $T_{BO}$. Recently, strong evidence of the emergence of the 1Q cLC state at ~130 K (>$T_{BO}$) has been reported by the magnetic torque measurement[21].

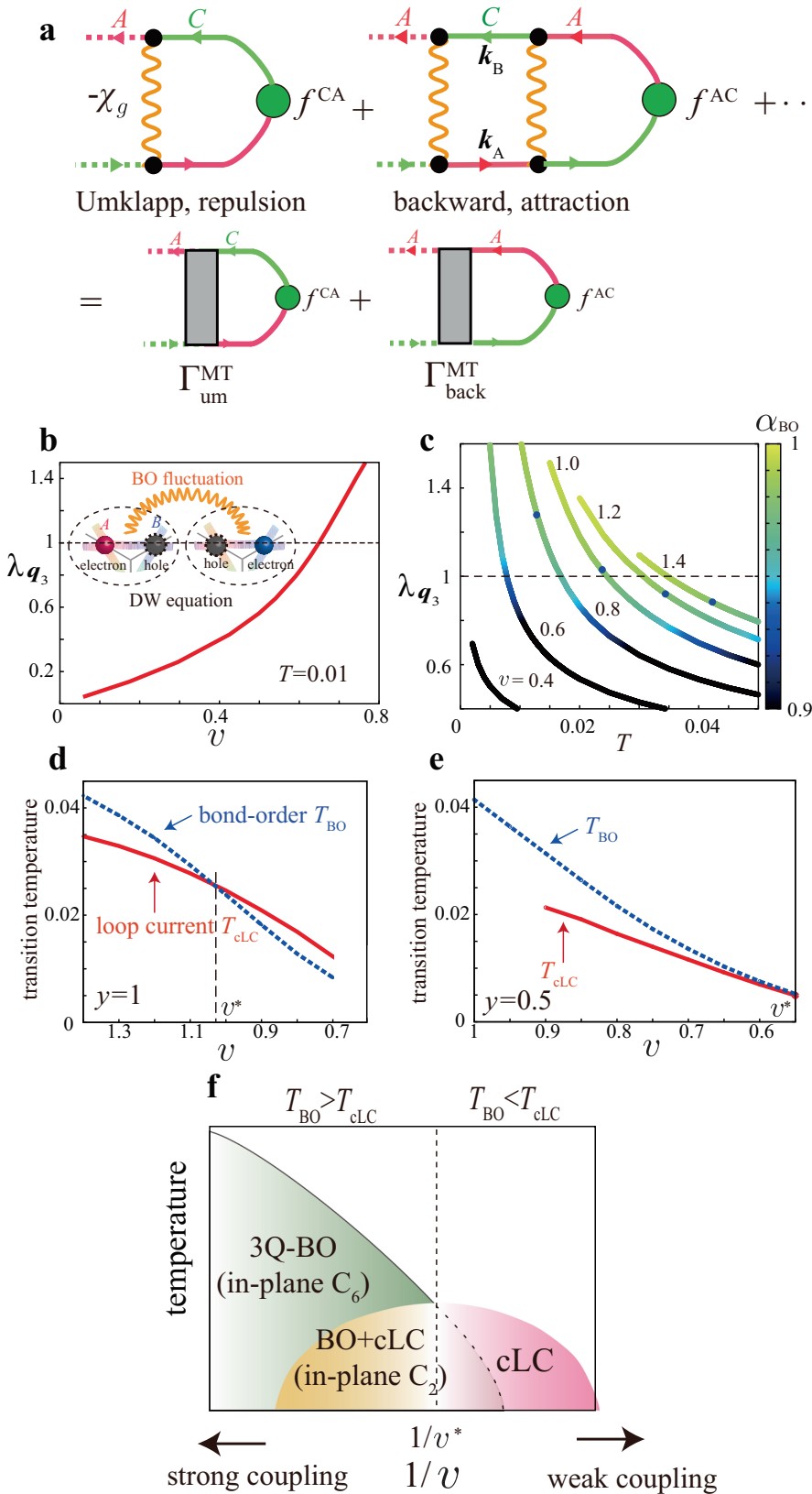

**Fig. 4 | cLC and BO transition temperatures and predicted phase diagram.**
**a** Series of Maki-Thompson (MT) processes produced in the density-wave (DW) equation. Yellow wavy lines represent the bond order (BO) propagators. The first-order and other odd-order terms give the repulsive Umklapp scattering $\Gamma^{MT}_{um} < 0$. The second-order and other even-order terms give the attractive backward scattering $\Gamma^{MT}_{back} > 0$. Both scatterings give the odd-parity charge loop current (cLC) order cooperatively. **b** Obtained $v$-dependence of the eigenvalue of cLC $\lambda_{\mathbf{q}_3}$ at $y = 1$. **c** Obtained $T$-dependence of $\lambda_{\mathbf{q}_3}$ at $y = 1$. $\alpha_{BO}$ is shown by the color of each line, and the small black circle on each line represents $T_{BO}$. The relation $T_{cLC} > T_{BO}$ is satisfied in the weak-coupling region ($v < v^*$). Obtained $T_{cLC}$ and $T_{BO}$ as functions of $v$ for **d** $y = 1$ and **e** $y = 0.5$. **f** Schematic phase diagram in the present theory. The nematic 3Q BO+cLC coexisting phase appears when $T_{cLC} < T_{BO}$.

The obtained nematic BO+cLC state is TRSB and two-dimensional. Other possible nematic state is the shift-stacking of the 3Q BO layers, each of which has $C_6$ symmetry. The shift-stacking is caused by the 3Q state composed of the 3D BO at $\mathbf{q}_n^{3D}$ with $q_{1,z}^{3D} = q_{2,z}^{3D} = \pi$ and $q_{3,z}^{3D} = 0$[10]. We stress that these two different nematic states can be realized at different temperatures.

## Giant AHE in cLC+BO state

Next, we discuss the transport phenomena that originate from the cLC[24,70]. Using the general expression of the intrinsic conductivity[65,71], we calculate the Hall conductivity ($\sigma_{xy}$ and $\sigma_{yx}$) due to the Fermi-surface contribution in the BO+cLC state. The expression is $\sigma_{\mu\nu} = \frac{1}{N}\sum_{\mathbf{k}} A_{\mu\nu}(\mathbf{k})$, where $A_{\mu\nu}(\mathbf{k}) = \frac{e^2}{\hbar}\frac{1}{\pi}\mathrm{Tr}\{\hat{v}_{\mathbf{k},\mu}\hat{G}_{\mathbf{k}}(i\gamma)\hat{v}_{\mathbf{k},\nu}\hat{G}_{\mathbf{k}}(-i\gamma)\}$. Here, $\hat{G}_{\mathbf{k}}(\epsilon) = ((\epsilon+\mu)\hat{1} - \hat{h}_{\mathbf{k}})^{-1}$ is the Green function matrix, where $\hat{h}_{\mathbf{k}}$ is the $12\times12$ tight-binding model with the 3Q BO and cLC order, and

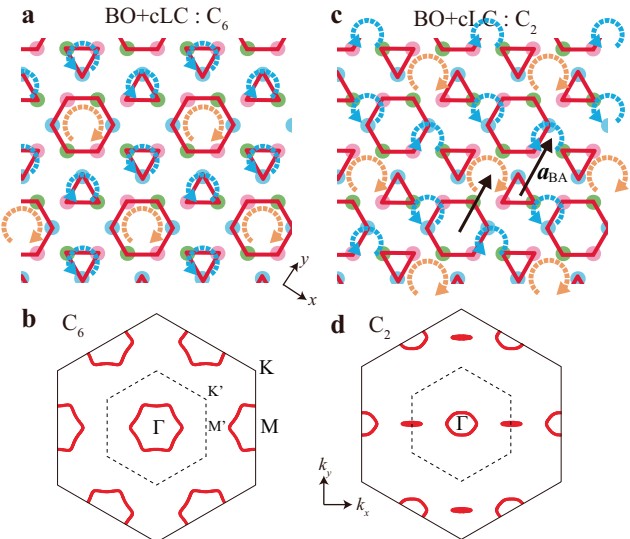

**Fig. 5 | cLC+BO coexisting states with $C_6$ and $C_2$ symmetries. a** $C_6$-symmetric bond order (BO) and charge loop current (cLC) coexisting state in real space. **b** Its folded Fermi surface (FS). The folded Brillouin zone is shown by dotted lines. **c** $C_2$-symmetric BO+cLC coexisting state. The nematicity originates from the out-of-phase combination of bond and current orders. **d** Its nematic FS. Here, the director is parallel to $\mathbf{a}_{BA}$ because the cLC order in (**c**) is shifted by $\mathbf{a}_{BA}$ from the cLC order in (**a**). Thus, the $Z_3$ nematic state with different three directors is realized. Here, we use large $|\delta t_{ij}^{b,c}|$ ( = 0.05) to exaggerate the nematicity.

$\hat{v}_{\mathbf{k},\mu} = d\hat{h}_{\mathbf{k}}/dk_\mu$ is the velocity operator. $\gamma$ ( > 0) is the electron damping rate that is given by the imaginary part of the self-energy. We set $n = n_{vHS}$ and $|\delta t_{ij}^{b}| = |\delta t_{ij}^{c}| = 0.025$, where the band hybridization gap due to the BO+cLC order is about $\Delta \approx 2\sqrt{|\delta t_{ij}^{b}|^2 + |\delta t_{ij}^{c}|^2} = 0.07$.

Figure 6a shows the obtained conductivities in the nematic BO+cLC state, in the unit of $\frac{e^2}{\hbar}$ ( = $2.4\times10^{-4}\Omega^{-1}$). When $\gamma \ll \Delta$, the Hall conductivity $\sigma_H \equiv \frac{1}{2}(\sigma_{xy} - \sigma_{yx})$ is almost constant, and its magnitude is proportional to $|\delta t_{ij}^{c}|$. When $\gamma \gg \Delta$, in contrast, $\sigma_H$ decreases with $\gamma$ in proportion to $\gamma^{-2}$. This crossover behavior is universal in the intrinsic Hall effect, which was first revealed in heavy fermion systems, and found to be universal in later studies[65,71–73]. Note that $\frac{1}{2}(\sigma_{xy} + \sigma_{yx})$ is nonzero in the nematic state. To understand the origin of the intrinsic Hall effect, we plot $A_H(\mathbf{k}) \equiv (A_{xy}(\mathbf{k}) - A_{yx}(\mathbf{k}))/2$ at $\gamma = 0.05$ in Fig. 6b: It shows a large positive value mainly around the vHS points, due to the band-hybridization induced by the cLC order. The obtained $\sigma_H \sim 1$ corresponds to $4\times10^3\Omega^{-1}\mathrm{cm}^{-1}$ because the interlayer spacing is ~0.6 nm. Thus, giant AHE $\sigma_H \sim 10^2\,\Omega^{-1}\mathrm{cm}^{-1}$ reported in refs. [25,26] is understood in this theory.

## Parquet RG theory, field-induced cLC mechanism

To verify the idea of the BO fluctuation-mediated cLC, we perform the analysis of the parquet renormalization group (RG) formulation[10,74] and present the results in Supplementary Note 5. A great merit of the RG method is that both particle-particle and particle-hole channels are treated on the same footing. We find that both BO and cLC fluctuations cooperatively develop in Supplementary Fig. 6. This result of the RG study strongly supports the validity of the DW equation analysis.

We comment on the complementary relationship between the present theory and the parquet RG theory. The latter theory solves a simplified 3-patch model in an unbiased way, leading to the development of both cLC order and BO, while the relationship between the two orders is not clear. On the other hand, the present theory focuses on the existence of the experimental BO phase and reveals that abundant BO quantum fluctuations lead to TRSB particle-hole condensation. Thus, the concept of the BO fluctuation-mediated cLC has been verified based on different reliable theories.

## Summary

In summary, we proposed a cLC mechanism mediated by the BO fluctuations in kagome metals. This cLC mechanism is universal because it is independent of the origin of the BO. The validity of the idea of the BO fluctuation-mediated cLC has been confirmed by the parquet RG study in Supplementary Note 5. Furthermore, we revealed

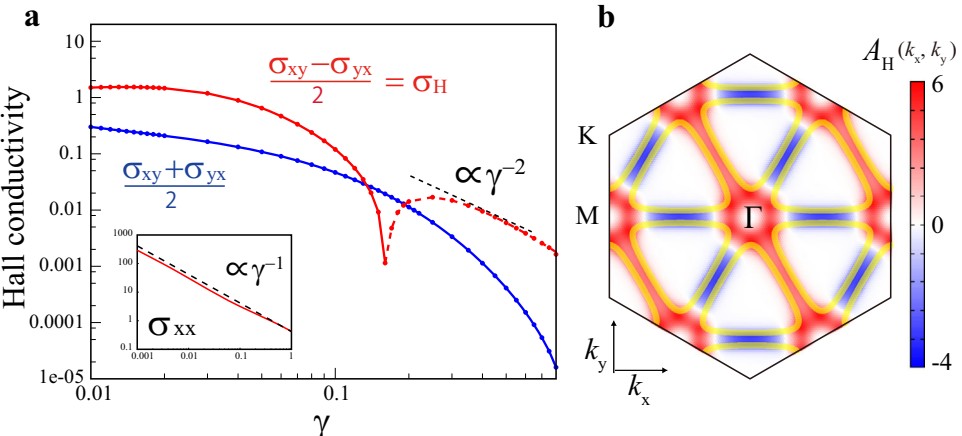

**Fig. 6 | Giant AHE in nematic cLC+BO state. a** Anomalous Hall conductivity in the nematic BO+cLC state ($|\delta t_{ij}^{b,c}| = 0.025$) as a function of the electron damping rate $\gamma \propto \tau^{-1}$. The full (broken) line represents the positive (negative) value. Thus, the Hall conductivity $\sigma_H$ becomes large in the low-resistivity region ($\gamma \lesssim 0.03$). **b** $A_H(\mathbf{k})$ at $\gamma = 0.05$: $\frac{1}{N}\sum_{\mathbf{k}} A_H(\mathbf{k}) = \sigma_H$.

that novel $Z_3$ nematicity emerges under the coexistence of the cLC and the BO reported in refs. 5,19,27 in addition to the giant AHE[25,26]. This theory presents a promising scenario for understanding the BO, the cLC and the nematicity in kagome metals in a unified way.

In the present study, we focus on the pure-type band composed of $b_{3g}$-orbital. However, the impact of other 3d-orbitals on the cLC order has also been studied in refs. 30,75. The extension of the present theory to multi-orbital models is a very important future issue.

Here, we shortly discuss several experimental evidences of the BO +cLC coexistence. Recent transport measurement of highly symmetric fabricated $CsV_3Sb_5$ micro sample[76] reveals that small magnetic field $h_z$ (<10T) or small strain gives rise to the nematic BO+cLC coexisting state below $T_{BO}$. This finding is well explained by the recent GL theory under $h_z$[77]: The current-bond-$h_z$ trilinear coupling caused by the orbital magnetization gives rise to the sizable $h_z$-induced cLC order in the BO state. This theory also explains $h_z$-induced enhancement of the cLC order observed by $\mu$SR measurements[15,17,18] and field-tuned chiral transport study[20]. It is noteworthy that the nematic electronic state that supports the $C_2$ BO+cLC order in Fig. 5c has been reported by recent STM measurement[69].

Finally, we comment on some interesting kagome metals other than $AV_3Sb_5$. Double-layer kagome metal $ScV_6Sn_6$ shows $\sqrt{3} \times \sqrt{3}$ charge-density-wave (CDW)[78]. It was proposed that the CDW originates from the flat phonon modes with Sn vibrations[79,80]. Interestingly, $ScV_6Sn_6$ also exhibits the spontaneous TRSB state[81]. The mechanism of the TRSB state in $ScV_6Sn_6$ is an interesting future problem. (Note that the existence of the vHS points is not a requirement for the cLC order[66]). The GL free energy analysis was performed in ref. 31. Recently, very weak but definite signal of the nematic electronic order has been observed in Ti-based kagome metal $CsTi_3Bi_5$[82,83]. To explain the observed hidden nematicity, the odd-parity BO without TRSB has been predicted theoretically[84].

## Methods

### Self-energy due to BO fluctuations

To understand the BO+cLC phase diagram and the energy scale of these orders accurately, we have to include the self-energy that describes the quasiparticle properties. We calculate the on-site self-energy due to BO fluctuations as

$$\Sigma_m(\epsilon_n) = \frac{T}{N} \sum_{\mathbf{k},q,m'',m'''} G_{m''m'''}(\mathbf{k}+\mathbf{q},\epsilon_n+\omega_l) \times B_{mm,m''m'''}(k,q), \tag{8}$$

$$B_{mm,m''m'''}(k,q) = g_{\mathbf{q}}^{m'm}(\mathbf{k}) g_{\mathbf{q}}^{m''m}(\mathbf{k})^* \cdot y v(1+v\chi_g(q)), \tag{9}$$

which is shown in Fig. 2d. Then, the Green function is given as $\hat{G}(k) = (i\epsilon_n + \mu - \hat{h}(\mathbf{k}) - \hat{\Sigma}(\epsilon_n))^{-1}$. The effect of thermal fluctuations described by the self-energy is essential to reproduce the $T$-dependence of various physical quantities. Here, $y = 1/2$ when $\hat{H}_{int}$ is given in Eq. (2). In the present numerical study, we calculate $\chi_g(q) = \chi_g^0(q)/(1 - v\chi_g^0(q))$ and $\Sigma_m(\epsilon_n)$ in Eq. (8) self-consistently.

### Kernel function of the DW equation

The kernel function due to BO fluctuations in Eq. (7) is given as

$$I_{\mathbf{q}}^{ll',mm'}(k,p) = - g_{\mathbf{p}-\mathbf{k}}^{m'l'}(\mathbf{k}) y v(1+v\chi_g(k-p)) g_{\mathbf{k}-\mathbf{p}}^{lm}(\mathbf{p}+\mathbf{q}) + g_{\mathbf{q}}^{ll'}(\mathbf{k}) v g_{\mathbf{q}}^{mm'}(\mathbf{p})^*, \tag{10}$$

which is expressed in Fig. 3a and Supplementary Fig. 4a. The first term, the MT term, is important when $\alpha_{BO} \lesssim 1$, and its first term is the Fock term. The second term, the Hartree term, vanishes when the eigenfunction $\hat{f}_{\mathbf{q}}(k)$ is orthogonal to the BO form factor $\hat{g}_{\mathbf{q}}(k)$, like the cLC

order is. Note that $\hat{B}(k,q) = -\hat{I}_0(k,k+q)$. A more detailed discussion is presented in Supplementary Note 3.

### Numerical analysis

In this study, we solved the eigenvalue equation with the kernel function (7) and the integral equations (8) and (9) numerically, by dividing the Brillouin zone into $60 \times 60$ **k** meshes. The number of **k** meshes is fine enough to achieve reliable numerical accuracy (~1%) at the calculated temperatures ($T \sim 0.01$).

## Data availability

The data generated by the present numerical study that is used in the figures of Main Manuscript (Figs. 2, 3, 4, 6) and those of Supplementary Information (Figs. 2, 4, 6, 8, 10, 11) has been deposited in the Figshare database.

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

## Acknowledgements

We are grateful to S. Onari, A. Ogawa, Y. Matsuda, T. Shibauchi, K. Hashimoto, and T. Asaba for fruitful discussions. This study has been supported by Grants-in-Aid for Scientific Research from MEXT of Japan (JP20K03858, JP20K22328, JP22K14003), and by the Quantum Liquid Crystal No. JP19H05825 KAKENHI on Innovative Areas from JSPS of Japan.

## Author contributions

R.T. executed the calculations in discussion with Y.Y and H.K., and R.T. and H.K. wrote the paper.

## Competing interests

The authors declare no competing interests.
