## [Peer Review File · Nature Communications]

REVIEWER COMMENTS

Reviewer #1 (Remarks to the Author):

In the manuscript “Charge-loop current order and Z3 nematicity mediated by bond-order fluctuations in kagome metal AV3Sb5 (A=Cs,Rb,K)” the authors present a theory for the emergence of a so-called charge loop current phase from bond-order fluctuations. The study is motivated by recent experiments on the AV3Sb5 kagome metals, which provide evidence for both time-reversal symmetry breaking and nematicity within the charge-ordered phase. The newly discovered AV3Sb5 compounds have been the subject of intense experimental and theoretical efforts attempting to understand the plethora of electronically ordered phases emerging at low temperatures. These include superconductivity and charge order, but also the above-mentioned more exotic phenomena. As a result, the manuscript presents a timely effort of interest to researchers working in the relatively narrow field of AV3Sb5 compounds, and it is certainly written with experts in mind (see comments below for details). However, the manuscript is of little relevance to researchers beyond this field.

The main result of the manuscript is presented as the calculation of the eigenvalue in Eq. (5). This equation was also solved in a previous paper (Ref. 13) involving the same authors. From what I can understand, the only distinction between the two works is a difference in the interaction mechanisms. By solving Eq. (5), the authors find that eigenvalues associated with a charge-loop current are found to cross unity first, thus signaling the formation of such an order. This provides a possible reason for the observation of time-reversal symmetry breaking in experiments. However, the technical details of the manuscript are presented in a very opaque manner and the generality of the result is hard to judge. Additionally, given the similarity to the results presented in Ref. 13, the novelty of the work is unclear (more on these comments below). Moreover, I have several comments and questions to the results and their interpretation, which the authors should address prior to the manuscript being considered for publication. These fall into two broad categories; one contains minor questions, comments, and suggestions, while the other category contains major questions or comments.

Major questions / comments:

1. Throughout the manuscript, the authors argue that their considerations are valid and relevant for the AV3Sb5 kagome metals. However, the electronic structure employed exhibits only one Fermi surface, in comparison to the three Fermi surfaces observed in ARPES experiments (e.g., Refs. 39, 41, 42, Nat. Phys. 18, 301, and PRL 128, 036402). While this corresponds to the Fermi surface of the so-called “pure” type associated to one of the bands giving rise to a van Hove singularity, I am not aware of any argument as to why this should be more important than the Fermi surface of the “mixed” type, also present (Ref. 39 and Nat. Phys. 18, 301). Furthermore, while there are heuristic

arguments that the Sb-dominated pocket near Gamma should be less important, this has, to my knowledge, yet to be confirmed by explicit calculations (references proving this statement wrong are welcome). Hence, there is no reason why including these additional bands in the electronic structure would not modify the results obtained in the manuscript. In the absence of a calculation including all three Fermi surfaces, the authors should make clear that the electronic structure applied has some deficiencies when compared to that of the AV3Sb5 compounds and soften the claims surrounding the implications for the kagome metals.

2. There are several aspects of the methods that I cannot follow. If I understand correctly, from the definition in Eq. (1), the indices l and m cannot be the same sublattice? By the definition of the operator O below Eq. (2) and the definition of the bond-order susceptibility in Eq. (3), this expression only contains bond fluctuations, is that correct? It is not clear if there is any such constraint on the indices from Eqs. (6) and (7) in the Methods section. This is important as a main point of the manuscript is that bond fluctuations drive the formation of the loop current order, so site fluctuations should not be present in this sum (or if they are, they should be demonstrated to be weak compared to the bond fluctuations). If the two are decoupled, the susceptibility of the site-order should be plotted to demonstrate that this is clearly subleading compared to the bond-order susceptibility. This issue is related to another question, the RPA susceptibility in Eq. (4). This is presented as an equation for a scalar quantity χ . However, in systems such as these where there are multiple degrees of freedom (orbitals, sites, etc.), the sum should not take place until after the RPA susceptibility has been evaluated, otherwise certain scattering channels are neglected. I think this is clearly seen in New. J. Phys. 11 025016 for the multi-orbital iron-based superconductors [Eqs. (19) and (20)]. Hence, should I understand Eq. (4) as a matrix equation, or has the sum already been carried out? If the sum over indices has already been carried out, what is the argument for doing so? Presumably this might lead to a completely different instability. In this regard, I think it would be very helpful for the reader if the authors showed $\chi_0(q)$ and contrasted it with the eigenvalues of the bare susceptibility not restricted to only bond fluctuations. This would illustrate the exact impact of the bond fluctuations and make their importance apparent to the reader.

3. I am confused about the normalization of g as stated below Eq. (2). If I assume that the bond order is at M then $q=M$, do I then understand correctly that the largest value of g as a function of l , m , and k is scaled to unity? What does this imply about the magnitude of the hopping element Δt_{ij} ? Since g enters explicitly in the susceptibility [Eq. (7)] which measures the strength of the bond-order fluctuations, I worry that such a rescaling will artificially promote the impact of the bond-order fluctuations when compared to the non-bond-order fluctuations. Is this true? I guess this concern is also related to the question above of the extent to which non-bond-order fluctuations have been accounted for in the calculations.

4. I am very puzzled by the claim that the so-called cLC+BO state breaks the rotational symmetry of the lattice. I am unable to reproduce the claims of a rotational symmetry-breaking phase when carrying out a brute-force numerical minimization of the full free energy (using NMinimize in

Mathematica with no assumptions imposed on the six order parameters). Whenever a phase with coexisting bond-order and loop-current order appears with order at all three q_n , this phase respects the rotational symmetry of the lattice. I have carried out this brute-force minimization for the free energy including the quartic terms which must be present to bound the free energy from below. For the regime $b_1=b_2$ quoted by the authors (and putting the quartic coefficients to 1) and assuming that the bond-order is leading, I find that the bond-order onsets first, followed by a reorganization at the onset of the loop-current in which the system breaks all remaining symmetries and make all six order parameters ($\phi_1, \phi_2, \phi_3, \eta_1, \eta_2, \eta_3$ in the notation of the authors) inequivalent. While I follow the arguments presented in the Supplementary Material Section C, it is clear from the numerical approach that the assumption that the three ϕ order parameters have identical magnitude, and the three η order parameters have identical magnitude are too restrictive and lead to wrong results. If my results are correct, it would imply that the phases found by the authors do not break rotational symmetry and should not be termed nematic. I therefore urge the authors to recheck their numerical minimization.

5. The relation between the current manuscript and Ref. 13 by the same authors is not fully clear to me. If I understand correctly, in Ref. 13 the authors demonstrate, by using a different kernel in Eq. (5), that (real) bond order forms in a system with an electronic structure very similar to the one studied in the current manuscript. In the current manuscript, it is the fluctuations of this bond order which are supposed to drive the formation of a loop-current order. However, with a different interaction kernel, what is the motivation for focusing on the bond-order fluctuations in the present manuscript? To demonstrate that the bond-order fluctuations are driving the formation of loop-current order in the first place, I think one should start from a scenario in which bond-order fluctuations are the strongest, as the one in Ref. 13. That being said, considering that the two interaction kernels are – at least to me – dissimilar, they produce very similar results for the real bond order [Fig. 2(c) of Ref. 13] and the imaginary bond order [Fig. 2(c) in the present manuscript]. On the other hand, the real bond-order described in Fig. 2(c) of Ref. 13 is not the one assumed present in the current manuscript [Fig. 1(d)]. What is the reason for the similarity between the real bond-order of Ref. 13 and the imaginary bond-order in the current manuscript? Moreover, the authors should make clear the reasoning why bond-order fluctuations are expected to be dominant, when the interactions are different from those stated in Ref. 13.

6. The authors employ a temperature which is 10% of the hopping integral t (as stated in the section “BO form factor and fluctuations”). This is quite far from the low-temperature regime in which these orders appear in the experiments, and I would expect to see significant smearing from the use of such a high temperature. Indeed, in the units provided, it corresponds to roughly 500 K (assuming 1 meV is 10 K which is a slight overestimate). The high temperature becomes entirely inappropriate when the authors claim that their results are consistent with the nematic order observed to onset near 35 K in CsV₃Sb₅ (in the section Z3-nematic state given by the cLC-BO coexistence). The calculations need to be carried out at a much lower temperature if they are to be relevant for the experimental observations. While I am aware that RPA frequently overestimates transition

temperatures substantially, I am not convinced that finite-temperature smearing has not had a significant impact on the results in this case.

7. The schematic phase diagram in Fig. 3(d) appears with no explanation of how the authors obtain it. Specifically, it is unclear from the manuscript how one goes from Fig. 2(b) and Fig. 3(c) to the schematic phase diagram. If I'm not mistaken, the calculations presented in the manuscript [specifically in Fig. 2(b)] are carried out in the disordered phase, i.e., the real bond order has not yet condensed (at least there is no mention of a Fermi surface reconstruction associated with such a condensation). The prominent bond-ordered dome in Fig. 3(d) leads the reader to believe otherwise. Furthermore, from Fig. 3(c) the leading eigenvalue is a strongly increasing function of v , in contrast to the dome-shape depicted in Fig. 3(d). Finally, as pointed out in Ref. 12, 3Q imaginary charge order (loop-current order) of the type studied here, cannot exist in the absence of a 3Q real charge order (bond order), as demonstrated by the presence of trilinear terms in Eq. (14) in Ref. 12 and stated explicitly in the first column of p. 10 of Ref. 12. The authors have the same trilinear terms in their free energy in Eq. (S20). As a result, the right-hand side of the schematic phase diagram cannot be realized for the orders studied.

Minor questions/comments:

m1. The kinetic Hamiltonian is never specified explicitly. I assume the authors are employing the standard one from, e.g., Eq. (2) of Phys. Rev. B 80, 113102 and that the filling factor quoted by the authors refers to total filling (spin up + spin down) and not filling per spin? Please clarify this. I think the authors should list the Hamiltonian in the methods section, so it is accessible to readers not familiar with the kagome lattice.

m2. In the introduction, the authors motivate their reasons for considering the emergence of a charge-current loop phase within the bond-ordered phase. However, much of the motivation is only valid for the CsV3Sb5 compound. Both the K- and Rb-variants exhibit time-reversal symmetry-breaking at the charge-order transition (see Refs. 16, 17, and arxiv:2202.07713). Even in CsV3Sb5, the temperature at which time-reversal symmetry is broken is controversial. Some references report it to be at the charge-order transition (e.g., Refs. 17 and Phys. Rev. Research 4, 02324) while others (e.g., Refs. 14 and 15) report it to be below the charge-order transition. These subtleties are important for the reader to know. It should also be reflected in the introduction that this is a rapidly progressing field, and many findings remain controversial. For instance, the observation of chiral charge order in Ref. 6 could not be reproduced in Ref 7.

m3. The authors should specify in the text (not in the supplementary) what the a -vector and the q_n -vectors are. This is known by experts but not by all readers.

m4. The authors should specify what is meant by B_{3g} orbitals in this case. While experts will know that this refers to the irrep of D_{2h} , the site-symmetry group of the vanadium site, it is unlikely that most readers will understand this. Moreover, the authors should explain what relevance the orbital character of their one-orbital model has.

m5. While experts will be aware of the Ginzburg-Landau theory referenced in the section “BO fluctuation-mediated cLC order” this might not be the case for all readers. Hence, for the reader to properly understand the meaning of $K_3/2K_4$, the authors should include the expression for the free energy involving these coefficients. Moreover, details of the calculation of K_3 and K_4 should be included. Does this calculation depend on the bond-order fluctuations and Maki-Thompson diagrams?

m6. The numerical model, introduced in Supplementary Material A, requires more explanation. The authors should demonstrate that this model does not violate the symmetries imposed by the kagome lattice, specifically, it should respect six-fold symmetry in the disordered phase. The authors should clarify the conditions imposed on the momentum-space vectors by this fact.

m7. The authors stress that the method employed goes beyond RPA (i.e., beyond mean-field theory). From the derivation of Eq. (S15) it was not clear to me how this differs from RPA with a long-ranged interaction $V(q)$. The contribution from the Maki-Thompson diagrams is stated in Eq. (S12) with no additional explanation for the reader to understand the origin. Moreover, Fig. S2(b) represents the eigenvalues of an equation that is never stated but, if I correctly, these are the solutions to the eigenvalue equation of Eq. (5) but with the interaction restricted to only Hartree-Fock terms? I think it would be very helpful to the reader if the authors made a similar plot for the interaction they employ in the main paper and include further details of the interaction kernel leading to Fig. S2(b). If the authors wish to make the claim that the method goes beyond RPA, it should be clear where these beyond-RPA contributions originate from, how the expressions are modified, and how the results are impacted.

m8. In Fig. S2, there are three labels in the figure yet four in the figure text. Is there a label missing on the bottom figure?

m9. I feel the list of references do not accurately reflect the rapid progress in both experimental and theoretical work in this field. On the experimental side, this should be partly clear from the number of times in the above questions or comments where I reference publications that are not part of the references in the manuscript. On the theoretical side, the authors cite 20 of their own theoretical papers, while citing 13 other theoretical works in total. Of these, only five are specific to the kagome

metals. While this could simply be a consequence of a very selective citation mentality, this seems contradicted by the 20 references to their own work.

In summary, I cannot recommend publication of the manuscript at the current stage. As is evident from the questions and comments above, there are potentially severe issues that need to be addressed before the manuscript can be reconsidered. Questions 1-6 of the in the “Major questions/comments” have the potential to impact the validity of the conclusions and it is crucial that the authors address these. Moreover, as also explained above, the manuscript is hard to follow, and the connection between figures and the results presented in the text is not always apparent.

Reviewer #2 (Remarks to the Author):

The manuscript by Hiroshi Kontani et al proposes a mechanism for charge loop current (cLC) order due to bond order(BO) fluctuations, and discusses the consequence of the coexisting phase to understand the nematicity and anomalous Hall effect. It is motivated by the recently extensively studied Kagome metals compounds AV_3Sb_5 ($A=K, Cs, Rb$), which show a cascade of quantum phases.

While the mechanism proposed in this work sounds interesting, I think the manuscript is not clear to follow (for one working in this field), and I do not fully understand and be convinced with the mechanism proposed in this manuscript based on the calculations presented. Therefore I cannot recommend the current version of the draft to publish in Nature Communications.

First, the notations of the manuscript should be clearer to follow. For example, the terms “linearized DW equation”, “MT process”, “K3 & K4” etc used in the text are not defined. There also seems to be some subfigure missed in Fig. S2. The author also mentioned a few times the “paramagnon interference mechanism”, I think it should be explained for general readers in the field of correlated electron systems.

Second, the discussions on why the cLC order can be stabilized by the BO fluctuations should be expanded. It would be very helpful if more mathematical details can be provided about the role of 1st order BO and 2nd order BO fluctuations to stabilize the cLC. In fact, I have a hard time to understand how this mechanism is consistent with the g-ology study presented in Ref. 12 by Park et al. The 1st order term (Umklapp) correspond to the g_3 in their Table I, 2nd order term correspond to the g_2 in their Table I, it seems to me the claim made in the manuscript on page 6 and caption of Fig.

3 is not consistent with their observation (which is both g_2 and g_3 should be repulsive to stabilize the cLC order)?

Third, to quantitatively support the key mechanism presented here, e.g. schematic phase diagram in Fig. 3d, I suggest the author to compute the transition temperature for the cLC near the quantum critical point of the BO, which can show if the dome for cLC really exist. Currently, from the results presented, I don't see what is the critical interaction v for the BO, and if the critical temperature is finite for the induced cLC at this critical v .

Reviewer #3 (Remarks to the Author):

Rina Tazai, Youichi Yamakawa, and Hiroshi Kontani report on the role of bond-order fluctuations for the charge-loop current order and Z3-nematicity in kagome metal AV_3Sb_5 ($A = Cs, Rb, K$). This is an interesting work addressing an important question about the interplay between charge-loop current and the bond-order. The paper will be of interest to both experimentalists and theorists. But before I can recommend the paper for publication, I have some comments/questions, shown below:

1. According to muon-spin rotation (μ SR) measurements the time-reversal symmetry breaking state seems to occur at the onset of charge order in all three compounds KV_3Sb_5 (Nature 602, 245-250 (2022)), RbV_3Sb_5 (<https://arxiv.org/abs/2202.07713>) and CsV_3Sb_5 (Phys. Rev. Research 4, 023244). This is also supported by Kerr effect measurements, as mentioned by the authors as well. It is indeed true that the magnetic response gets stronger at some lower temperature. For instance, in RbV_3Sb_5 very well pronounced two-step increase of the magnetic relaxation rate was reported by μ SR with two characteristic temperatures of $T_1^* = 110$ K (onset temperature for charge order) and $T_2^* \approx 50$ K (<https://arxiv.org/abs/2202.07713>). In the same work, authors followed the TRSB charge order transitions in RbV_3Sb_5 as a function of pressure. The obtained phase diagram shows that the two-step transition becomes a single time-reversal symmetry-breaking transition at 1.58 GPa, above which $T_1^* = T_2^*$ shows a faster suppression and follows the phase boundary of the charge order. Two transitions with $T_1^* = 90$ K and $T_2^* \approx 30$ K are also observed in CsV_3Sb_5 . This shows that time-reversal symmetry breaking signal appears right at the onset of charge order and gets enhanced below 50 K in RbV_3Sb_5 and below 30 K in CsV_3Sb_5 . How do these experimental observations fit into the picture of bond-order fluctuations mediated loop current? Authors should extend the discussion on this aspect and more clearly discuss the connection between the predicted phase diagram shown in Figure 3d and the experimental observations.

2. From the experimental front, at ambient pressure a nodeless SC gap is reported for CsV_3Sb_5 and nodal SC pairing for RbV_3Sb_5 and KV_3Sb_5 (<https://arxiv.org/abs/2202.07713>). But once charge order

is either strongly suppressed or fully suppressed, the nodeless state is stabilised for all three compounds. So, it seems that depending on the fine details of the charge ordered state in AV₃Sb₅, either nodal or nodeless state can be found. It would be helpful if authors can comment on the superconducting gap symmetry in these kagome materials from the perspective of bond-order fluctuations theory. Is there a parameter space of charge order in the theoretical model in which a nodal gap is stabilized?

Reply to Reviewer #1:

○ Comments 1-1:

In the manuscript “Charge-loop current order and Z3 nematicity mediated by bond-order fluctuations in kagome metal AV₃Sb₅ (A=Cs,Rb,K)” the authors present a theory for the emergence of a so-called charge loop current phase from bond-order fluctuations. The study is motivated by recent experiments on the AV₃Sb₅ kagome metals, which provide evidence for both time-reversal symmetry breaking and nematicity within the charge-ordered phase. The newly discovered AV₃Sb₅ compounds have been the subject of intense experimental and theoretical efforts attempting to understand the plethora of electronically ordered phases emerging at low temperatures. These include superconductivity and charge order, but also the above-mentioned more exotic phenomena. As a result, the manuscript presents a timely effort of interest to researchers working in the relatively narrow field of AV₃Sb₅ compounds, and it is certainly written with experts in mind (see comments below for details). However, the manuscript is of little relevance to researchers beyond this field.

○ Reply 1-1:

General and broad interest of the present study:

We are grateful to Reviewer #1 for his/her useful comments on the previous manuscript. We agree that AV₃Sb₅ has been the subject of “intense experimental and theoretical efforts attempting to understand the plethora of electronically ordered phases emerging at low temperatures.” The current most significant issue is the “spontaneous charge loop current (cLC) order with time-reversal-symmetry-breaking (TRSB)”, while its driving force is totally unknown. To solve this hot issue, we provide a promising novel idea of the “bond-order (BO) fluctuation mediated current order” mechanism. This mechanism is not restricted to BO fluctuations, so this idea will be useful to understand the current order in other metals, such as cuprate superconductors and iridates. We intensively revised the manuscript to attract the researchers in the wide field of condensed matter physics.

○ Comments 1-2:

The main result of the manuscript is presented as the calculation of the eigenvalue in Eq. (5). This equation was also solved in a previous paper (Ref. 13) involving the same authors. From what I can understand, the only distinction between the two works is a difference in the interaction mechanisms. By solving Eq. (5), the authors find that eigenvalues associated with a charge-loop current are found to cross unity first, thus signaling the formation of such an order. This provides a possible reason for the observation of time-reversal symmetry breaking in experiments. However, the technical details of the manuscript are presented in a very opaque manner and the generality of the result is hard to judge. Additionally, given the similarity to the results presented in Ref. 13, the novelty of the work is unclear (more on these comments below). Moreover, I have several comments and questions to the results and their interpretation, which the authors should address prior to the manuscript being considered for publication. These fall into two broad categories; one contains minor questions, comments, and suggestions, while the other category contains major questions or comments.

○ Reply 1-2:

“Beyond-RPA effect” in the present theory:

To clarify the point of the discussion, first of all, we would like to reply to Comment 1-16 by Reviewer #1, “If the authors wish to make the claim that the method goes beyond the random-phase-approximation (RPA), it should be clear where these beyond-RPA contributions originate from, how the expressions are modified, and how the results are impacted.” (on page 22): The “beyond-RPA contributions” that we study in the present theory are shown in **Fig S4 e** of the revised manuscript as well as in **Figs. R3 (d)** of the **SI of Author Reply B**. The cLC order originates from the Maki-Thompson (MT)-type diagrams that are “beyond-RPA”.

Universality of the DW equation method:

To answer the “major questions” by Reviewer #1, we would first explain that the “DW equation” provides the general and useful formalism for general quantum phase transitions in metals [44]. The phase transitions in metals, not only CDW and SDW but also bond and current orders, are described as the symmetry breaking of the normal self-

energy: $\Delta\Sigma \equiv \Sigma - \Sigma_{A1g}$ [44,50]. $\Delta\Sigma$ is determined by the stationary condition of the free energy $\delta F[\Delta\Sigma]/\delta(\Delta\Sigma) = 0$. This condition is rigorously satisfied by the solution of “the DW equation”, as proved based on the Luttinger-Ward theory in Ref. [50]. Because the DW equation is formally exact, it is applicable to both odd-parity BO and even-parity cLC.

The expression of the DW equation is $\lambda f = \int I(GG) f$, where I is interaction between the particle and hole, and $f \propto \Delta\Sigma$ is called the form factor. Various non s-wave form factor $f(\mathbf{k})$ can emerge when the interaction I is non-local. In this paper, we obtain the TRSB odd-parity solution $f(\mathbf{k}) \sim \sin k_x$, which is called the current order. More detailed explanations of the DW equation and the form factor are summarized in our recent review article [44].

(Note that the DW equation is analogous to the superconducting (SC) gap function: $\lambda\Delta(\mathbf{k}) = \int I |G|^2 \Delta(\mathbf{k})$, which is also derived from the stationary condition $\delta F[\Delta]/\delta\Delta(\mathbf{k}) = 0$.)

How to find reliable interaction I ?

The DW equation is formally exact, and we can construct reliable theory by choosing reliable I . It gives the random-phase approximation (RPA) if I is the lowest Hartree-Fock terms of H_{int} . When $H_{\text{int}} = U n_{i\uparrow} n_{i\downarrow}$, the RPA gives large spin susceptibility $\chi_s(q)$. Then, it is well established that the spin-fluctuation-mediated interaction $I \sim -U^2 \chi_s(q)$ drives various symmetry breaking states, like the d-wave superconductivity $\Delta(\mathbf{k}) \sim \cos k_x - \cos k_y$ [D. J. Scalapino et al., Phys. Rev. B 34, 8190(R) (1986)] and the d-wave bond order $f(\mathbf{k}) \sim \cos k_x - \cos k_y$ [44].

Reliability of the BO fluctuation-mediated cLC mechanism:

The BO interaction in **Eq. (2)** gives large BO susceptibility $\chi_g(q)$. Then, the strong interaction $I \sim -v^2 \chi_g(q)$, which is beyond-RPA, should play important roles. (This is a reasonable and convincing idea based on the success of spin fluctuation theories [63-65].) Based on this idea, we discovered the “BO fluctuation mediated current order” for the first time. This is a natural idea for kagome metals because the existence of strong BO fluctuations at $T^* \sim 160\text{K}$ has been reported by X-ray study [48] and thermal transport measurement [49]. This naturalness of the present loop current order theory will attract great attention.

Difference between the present theory and Ref. [11] (= [13] in the previous manuscript).

The reason for the different form factors given by the present study and Ref. [11] is

the “different functional forms of I in the DW equation. The cLC order in the present oduprtheory originates from the strong “repulsive Umklapp scattering” that is induced by the BO fluctuations (see **Fig. 4 a**). This effect is totally dropped in Ref. [11].

To seek the idea of “BO fluctuation-mediated cLC”, we introduce H_{int} in **Eq. (2)** instead of specifying the origin of the BO. This strategy is standard in condensed matter physics; It is analogous to the “spin fermion models” $H_{\text{int}} \propto \int dq \chi_0^{-1}(q) \mathbf{S}_q \cdot \mathbf{S}_{-q}$ for nearly AFM metals; see A. Abanov, A.V. Chubukov, and J. Schmalian, *Adv. Phys.* **52**, 119 (2003). We are confident that the idea of “BO fluctuation-mediated cLC” is crucial for kagome metals, and this concept should attract broad interest from readers of Nature Communications.

Change in the manuscript:

We added simple and useful explanations about the key idea to the revised manuscript as follows. We are confident that the present manuscript becomes much more readable for readers in a wide range of research fields.

(i) On the density-wave (DW) equation: (page 3)

“The phase transitions in metals are described as the symmetry breaking of the normal self-energy; $\Delta\Sigma \equiv \Sigma - \Sigma_{\text{Alg}}$ [44,50]. $\Delta\Sigma$ is determined by the stationary condition of the free energy; $\delta F[\Delta\Sigma]/\delta(\Delta\Sigma) = 0$. The DW equations enables us to derive the solution that satisfies the stationary condition, as we proved based on the Luttinger-Ward theory [50]. Based on the DW equation, we discover that the odd-parity and TRSB $\Delta\Sigma$ is driven by the BO fluctuation exchange processes. (Note that the DW equation for $\Delta\Sigma$ is analogous to the Eliashberg equation for the superconducting (SC) gap Δ .)”

(ii) On the Maki-Thompson (MT) term: (page 6)

“The MT term describes the scattering of electrons due to the developed bosonic fluctuations. This scattering process is important in metals near the quantum critical points. For example, in nearly antiferromagnetic metals, the d-wave SC transition is induced by the MT processes of spin fluctuations. In kagome metals, the MT term represents the strong inter vHS scattering of electrons mediated by the abundant BO fluctuations; see Fig. 3 a. (The MT term also describes the s -wave SC state in kagome metals [11].) Here, we find that both $\tilde{\chi}_g$ and $\tilde{\chi}_{\text{cLC}}$ are comparably enlarged due to the MT

processes in the present theory.”

○ Comments 1-3:

Major questions / comments:

1. Throughout the manuscript, the authors argue that their considerations are valid and relevant for the AV3Sb5 kagome metals. However, the electronic structure employed exhibits only one Fermi surface, in comparison to the three Fermi surfaces observed in ARPES experiments (e.g., Refs. 39, 41, 42, Nat. Phys. 18, 301, and PRL 128, 036402). While this corresponds to the Fermi surface of the so-called “pure” type associated to one of the bands giving rise to a van Hove singularity, I am not aware of any argument as to why this should be more important than the Fermi surface of the “mixed” type, also present (Ref. 39 and Nat. Phys. 18, 301). Furthermore, while there are heuristic arguments that the Sb-dominated pocket near Gamma should be less important, this has, to my knowledge, yet to be confirmed by explicit calculations (references proving this statement wrong are welcome). Hence, there is no reason why including these additional bands in the electronic structure would not modify the results obtained in the manuscript. In the absence of a calculation including all three Fermi surfaces, the authors should make clear that the electronic structure applied has some deficiencies when compared to that of the AV3Sb5 compounds and soften the claims surrounding the implications for the kagome metals.

○ Reply 1-3:

Thank you very much for the comment on the existence of the “mixed-type” Fermi surface (FS) and the p-orbital FSs in kagome metal. The former is composed of a d-orbital of V ion, so strong electron correlation exists. Regardless of this fact, the “pure type” FS plays the dominant contribution for the current order. The reason is that the nesting between the different sublattices (A,B), (B,C), (C,A), which drives the inter-sublattice BO (**Fig. 1 b**) and cLC (**Fig. 3 d**), is relevant only in the pure type FS. This is the reason why we concentrate on the pure type FS in the present theory.

Simple 3-site model is enough to investigate the BO fluctuation mechanism.

We have already studied the kagome metal model with both the “pure type” and the “mixed type” FSs in Ref. [11]. It was revealed that the BO emerges only on the “pure type” FS composed of a single d-orbital. This is a natural result because the “inter-sublattice (A-B) nesting at $\mathbf{q} = \mathbf{q}_1$ ” gives the experimental BO along y axis; see **Fig. 1**. Based on this finding, we decided to study the present simple 3-site model with “pure type” FS. The present 3-site model is very suitable for investigating the novel idea of “the cLC mediated by BO fluctuations that emerge on the pure type FS selectively”.

In the revised manuscript, we introduce the “next-nearest hopping integral $t' = -0.08$ ” to make the shape of the Fermi surface more realistic, as we show in **Fig. 1 c**. We find that the cLC order is robustly obtained for $|t'| \lesssim 0.10$. For a future quantitative study, analysis of a more realistic multiband model would be useful.

Change in the manuscript:

We added the following explanation to the revised manuscript on pages 4:

“We set $t (= -0.5\text{eV})$ to fit the bandwidth, and $t' (= -0.08\text{eV})$ to reproduce the shape of the Fermi surface (FS). Numerical results are insensitive to the presence of t' . Hereafter, the unit of energy is eV unless otherwise noted. The FS around the van-Hove singular (vHS) point ($\mathbf{k} \approx \mathbf{k}_A, \mathbf{k}_B$ or \mathbf{k}_C) is composed of a single 3d-orbital on V ion. $\dots \dots$. This simple three-site model well captures the main pure-type FS in kagome metals [4,52-56]. The wavevectors of the BO correspond to the inter-sublattice nesting vectors \mathbf{q}_n ($n=1,2,3$) in **Fig. 1 c**. $\dots \dots$. The good inter-sublattice nesting of the FS naturally triggers the observed inter-sublattice BO at $\mathbf{q} = \mathbf{q}_n$, as shown in previous theoretical studies [6,7,11].”

○ Comments 1-4:

2. There are several aspects of the methods that I cannot follow. If I understand correctly, from the definition in Eq. (1), the indices l and m cannot be the same sublattice? By the definition of the operator O below Eq. (2) and the definition of the bond-order susceptibility in Eq. (3), this expression only contains bond fluctuations, is that correct? It is not clear if there is any such constraint on the indices from Eqs. (6) and (7) in the Methods section. This is important as a main point of the manuscript is that bond

fluctuations drive the formation of the loop current order, so site fluctuations should not be present in this sum (or if they are, they should be demonstrated to be weak compared to the bond fluctuations). If the two are decoupled, the susceptibility of the site-order should be plotted to demonstrate that this is clearly subleading compared to the bond-order susceptibility. This issue is related to another question, the RPA susceptibility in Eq. (4). This is presented as an equation for a scalar quantity χ . However, in systems such as these where there are multiple degrees of freedom (orbitals, sites, etc.), the sum should not take place until after the RPA susceptibility has been evaluated, otherwise certain scattering channels are neglected. I think this is clearly seen in New. J. Phys. 11 025016 for the multi-orbital iron-based superconductors [Eqs. (19) and (20)]. Hence, should I understand Eq. (4) as a matrix equation, or has the sum already been carried out? If the sum over indices has already been carried out, what is the argument for doing so? Presumably this might lead to a completely different instability. In this regard, I think it would be very helpful for the reader if the authors showed $\chi_0(q)$ and contrasted it with the eigenvalues of the bare susceptibility not restricted to only bond fluctuations. This would illustrate the exact impact of the bond fluctuations and make their importance apparent to the reader.

○ Reply 1-4:

We would like to explain the **correctness of Eq.(4), Eq.(5), and Eq.(6)** (= Eq.(6), Eq.(7), and Eq.(4) in the previous manuscript) in the case of $H_{\text{int}} = \int (-v/2) O^g_q O^g_{-q}$ in **Eq. (2)** step by step. In **(i)-(iii)**, we explain that only the BO fluctuations are enlarged in the Hartree-Fock (HF) approximation. In **(iv)-(v)**, we reveal that the cLC fluctuations are as strong as the BO fluctuations owing to the beyond-RPA effect.

(i) “Absence of on-site terms” in $\chi_g^{0,lmml'}$ in Eq.(5):

As pointed out by Reviewer #1, the operator O^g (and the BO form factor g) is composed of only inter-site components, and intra-site components are absent. That is, $\chi_g^{0,lmml'}$ is zero when $l=m$ or $l'=m'$. Therefore, the BO susceptibility defined in **Eq. (3)** is exactly given by $\chi_g^0 = \sum_{l'mm'} \chi_g^{0,lmml'}$ in **Eqs. (4) and (5)** of the revised manuscript.

(ii) BO susceptibility χ_g is dominant in the Hartree approximation:

In the presence of H_{int} , the BO susceptibility is exactly given as $\chi_g^{\text{RPA}} = \chi_g^0 + \chi_g^0 v \chi_g^0 + \chi_g^0 v \chi_g^0 v \chi_g^0 + \dots = \chi_g^0 / (1 - v \chi_g^0)$ in the Hartree approximation, as we show in **Fig. 2 c** in the revised manuscript. It diverges when $\alpha_{\text{BO}} \equiv v \chi_g^0$ is unity. In the “**SI of Author Reply: A** (on page 41)”, we derive the same scalar χ_g^{RPA} in terms of the “matrix equation” to reply to the concern by Reviewer #1. (It is shown that $\det(\hat{1} - \hat{V} \hat{\chi}_g^0)$ is exactly zero when $\alpha_{\text{BO}} = 1$, where \hat{V} is the matrix expression of the BO interaction H_{int} .)

For susceptibility of a general form factor w , χ_w^{RPA} , is given in **Figs. R3 (a) and (b)** in the “**SI of Author Reply: B** (on page 42)”. It is found that $\chi_w^{\text{RPA}} = \chi_w^0$ when $w = \text{”onsite”}$ form factor (such as $w_{\text{im}} \sim \delta_{i,m}$). The reason is that $\chi_{w,g}^0 \equiv -T \sum w(GG)g$ introduced in Fig. R3 (b) is almost zero when $w = \text{”onsite”}$ because the two form factors g and w are orthogonal. For this reason, $\chi_g^{\text{RPA}} \gg \chi_w^{\text{RPA}}$ when $\alpha_{\text{BO}} \lesssim 1$. Therefore, the main concern by Reviewer #1 is resolved.

(iii) χ_g is dominant in the Hartree-Fock (HF) approximation:

We verify the relation $\chi_g^{\text{RPA}} \gg \chi_w^{\text{RPA}}$ ($w \neq g$) in the HF approximation in the revised **SI C: Figure S4 b** shows the eigenvalues of the DW equation as functions of v . In the region $v < 0.5$, which corresponds to the HF result, the largest and the second largest eigenvalues corresponds to the BO and cLC, respectively. The eigenvalues of “onsite orders” are further smaller. Therefore, χ_w^{RPA} is the largest for $w = g$.

(iv) cLC susceptibility becomes as large as χ_g owing to beyond-RPA effect.

We find that the cLC instability is magnified as large as the BO instability owing to the beyond-RPA effect. In **Fig. S4 b**, in the region $v \gtrsim 1$, the cLC instability is strongly enlarged by the MT term that is proportional to $(1 - v \chi_g^0)^{-1} \gg 1$. Similar results are obtained in **Figs. 4 d-e** in the main text. (The origin of the cLC is the “repulsive Umklapp” and “attractive backward” interactions driven by the MT terms shown in **Fig. 4 a**.) Thus, the present study leads to the novel concept “BO fluctuation mediated cLC mechanism”, which is the main finding of the present study.

(v) parquet RG analysis: beyond-RPA theory

Furthermore, in the revised **IS F**, we present another powerful verification of the cLC

mechanism based on the “parquet RG theory”, which is reliable and unbiased beyond-RPA theory. The obtained cooperative development of the cLC and BO fluctuations in **Fig. S11 d** and **f** are presented in the result in the revised **SI F**.

In the parquet RG theory, the instability for the “onsite operator $n_A + s n_B$ ” ($s = \pm 1$) at $\mathbf{q} = \mathbf{q}_1$ is $G_{CDW,s} = s(-g_1 + 2g_2) - g_4$, and that for the “onsite operator $m_A + s m_B$ ” ($m_A = n_{A\uparrow} + n_{A\downarrow}$) at $\mathbf{q} = \mathbf{q}_1$ is $G_{SDW,s} = s g_1 + g_4$. Both quantities are smaller than G_{cBO} and G_{cLC} after the renormalization, as verified in **Fig. S11 c - f**. Therefore, the “site fluctuations” are not relevant in the present model based on two theoretical frameworks. Thus, the main concern by Reviewer #1 is resolved.

Change in the manuscript:

We added the following explanation to the revised manuscript on page 5:

“Equation (5) contains two form factors, so it vanishes when $l=m$ or $l'=m'$. Its diagrammatic expression is given in Fig. 2 a. The numerical result for $\chi_s^{0,ABAB}(\mathbf{q},0)$ is shown in Fig. 2 b, which exhibits the broad peak at the nesting vector between vHS-A and vHS-B; $\mathbf{q} = \mathbf{q}_1$.”

“The BO susceptibility in Eq. (3) is strongly magnified by the Hartree term of the Eq. (2) because of the same form factors in both equations. Its process is expressed in Fig. 2 c,”

○ Comments 1-5:

3. I am confused about the normalization of g as stated below Eq. (2). If I assume that the bond order is at M then $q=M$, do I then understand correctly that the largest value of g as a function of l , m , and k is scaled to unity? What does this imply about the magnitude of the hopping element $\forall \delta \Delta t_{ij}$? Since g enters explicitly in the susceptibility [Eq. (7)] which measures the strength of the bond-order fluctuations, I worry that such a rescaling will artificially promote the impact of the bond-order fluctuations when compared to the non-bond-order fluctuations. Is this true? I guess this concern is also related to the question above of the extent to which non-bond-order fluctuations have been accounted for in the calculations.

○ Reply 1-5:

Normalization of form factor does not change the results:

First, we explain that the present normalization rule for g and f is convenient and physically reasonable. In our study, the BO and cLC order parameters at $\mathbf{q}=\mathbf{q}_l$ are $\phi_l g_q^{lm}(\mathbf{k})$ and $\eta_l f_q^{lm}(\mathbf{k})$ with $(l,m)=(A,B)$ or (B,A) , respectively. The unit of ϕ_l and η_l is eV, and we have to introduce a “normalization rule of the dimensionless form factors g and f ” to exclude to arbitrariness. Here, we normalize them as $\max_{\mathbf{k}} |g_{q_l}^{BA}(\mathbf{k})| = \max_{\mathbf{k}} |f_{q_l}^{BA}(\mathbf{k})| = 1$, which is realized at $\mathbf{k}=\mathbf{k}_A$ in the present DW equation solutions. Then, the BO $\phi g_q(\mathbf{k})$ [cLC order $\eta f_q(\mathbf{k})$] gives the “hybridization gap $\Delta_{BO} \sim |\phi|$ [$\Delta_{cLC} \sim |\eta|$]” in the folded band at Γ point. Then, the kinetic energy gain due to the band hybridization is directly given as $\sim -N(0)|\phi|^2$ [$\sim -N(0)|\eta|^2$] for the BO [cLC] formation, where $N(0)$ is the density-of-states at Fermi level.

Next, we explain that “**the normalization rule does not influence the physical quantities**”. The reason is that the change of H_{int} in Eq. (2) due to $g \rightarrow gr$ is absorbed by $v \rightarrow v/r^2$. Both g and v appear only in the interaction H_{int} . Thus, the following concern made by Reviewer #1 is eliminated: “I worry that such a rescaling will artificially promote the impact of the bond-order fluctuations when compared to the non-bond-order fluctuations. Is this true?”

Also, the GL phase diagram with C_2 coexisting phase in Fig. S10 in the revised SI E-2 is unchanged. (η and ϕ in Fig. S10 b and c are just multiplied by r .)

Change in the manuscript:

We added the following explanation to the revised manuscript in the SI E-2:

“We normalize them as $\max_{\mathbf{k}} |g_{q_l}^{BA}(\mathbf{k})| = \max_{\mathbf{k}} |f_{q_l}^{BA}(\mathbf{k})| = 1$, which is realized at $\mathbf{k}=\mathbf{k}_A$ in the present DW equation analysis. Then, the BO parameter $\phi_m g_{q_m}(\mathbf{k})$ at $\mathbf{q}=\mathbf{q}_m$ ($m=1\sim 3$) gives the hybridization gap $\Delta_{BO} \sim |\phi|$ in the folded band at a Γ point. [In the same way, [$\Delta_{cLC} \sim |\eta|$] for the cLC order $\eta_m f_{q_m}(\mathbf{k})$.] For this reason, the present normalization rule for g and f is physically reasonable and convenient. Note that the normalization rule does not influence the physical quantities, because the change of H_{int} in Eq. (2) due to $g \rightarrow gr$ is absorbed by $v \rightarrow v/r^2$.”

○ Comments 1-6:

4. I am very puzzled by the claim that the so-called cLC+BO state breaks the rotational symmetry of the lattice. I am unable to reproduce the claims of a rotational symmetry-breaking phase when carrying out a brute-force numerical minimization of the full free energy (using NMinimize in Mathematica with no assumptions imposed on the six order parameters). Whenever a phase with coexisting bond-order and loop-current order appears with order at all three q_n , this phase respects the rotational symmetry of the lattice. I have carried out this brute-force minimization for the free energy including the quartic terms which must be present to bound the free energy from below. For the regime $b_1=-b_2$ quoted by the authors (and putting the quartic coefficients to 1) and assuming that the bond-order is leading, I find that the bond-order onsets first, followed by a reorganization at the onset of the loop-current in which the system breaks all remaining symmetries and make all six order parameters ($\phi_1, \phi_2, \phi_3, \eta_1, \eta_2, \eta_3$ in the notation of the authors) inequivalent. While I follow the arguments presented in the Supplementary Material Section C, it is clear from the numerical approach that the assumption that the three ϕ order parameters have identical magnitude, and the three η order parameters have identical magnitude are too restrictive and lead to wrong results. If my results are correct, it would imply that the phases found by the authors do not break rotational symmetry and should not be termed nematic. I therefore urge the authors to recheck their numerical minimization.

○ Reply 1-6:

In the revised **SI E-2**, we examine the realization condition for the C_2 coexisting state in much more detail by “minimizing the GL free energy $F[\phi, \eta]$ exactly with respect to ϕ and η ”. We study the following 13^2 patterns $(\phi, \eta) = (\phi_m, \eta_n)$ with $m, n = 1 \sim 13$:

$$3Q \text{ BO: } \phi_1 = \phi(1,1,1) / \sqrt{3},$$

$$\phi_2 = \phi(-1,1,1) / \sqrt{3}, \quad \phi_3, \phi_4 = \text{cycl.},$$

$$2Q \text{ BO: } \phi_5 = \phi(1,1,0) / \sqrt{2}, \quad \phi_6, \phi_7 = \text{cycl.},$$

$$\phi_8 = \phi(1,-1,0) / \sqrt{2}, \quad \phi_9, \phi_{10} = \text{cycl.},$$

$$1Q \text{ BO: } \phi_{11} = \phi(1,0,0), \quad \phi_{12}, \phi_{13} = \text{cycl.},$$

1Q~3Q CLC: $\eta_m = \phi_m|_{\phi \rightarrow \eta}$ ($m=1\sim 13$).

Thanks to the present numerical study, the C_2 BO+cLC coexisting phase is realized in the purple region in **Fig. S10 a**. Here, the GL coefficients are derived based on the diagrammatic method. **Figure S10 a** means that the C_2 coexisting phase is always realized for $T_{\text{cLC}} \lesssim T_{\text{BO}}$. Below, we present more detailed explanations.

Violation of the ideal relations of GL coefficients in real systems:

It is known that the C_2 coexisting state is prohibited when two ideal equality relations $d_{1a}=d_{2a}=d_{3a}$ and $d_{1b}=d_{2b}=d_{3b}$ in Eq. (S28) are satisfied, as proved in Ref. [10] and pointed out by Reviewer #1. Here, d_{1a} , d_{2a} , d_{3a} , d_{1b} , d_{2b} , d_{3b} are 4th order GL coefficients. However, the C_2 coexisting state can emerge when these ideal relations are violated (in case d_{3a} and d_{3b} are relatively small), as we have already pointed out in the **previous SI D**.

In the revised **SI E-2**, we calculate all the GL coefficients to perform the GL analysis. We use the cLC form factor derived from the DW equation analysis in the main text. All the obtained 4th order coefficients are different because the \mathbf{k} - and ε_n -dependences of the cLC form factor are very different from those of the BO form factor; see **Fig. S7**. (This effect cannot be discussed by the parquet RG method in the **SM F**.)

As we show in **Fig. S9 d**, the obtained ratios $R=d_{1a} \cdot d_{2a}/(d_{3a})^2$ and $R'=d_{1b} \cdot d_{2b}/(d_{3b})^2$ are larger than unity. Therefore, the cLC-BO competing terms, d_{3a} and d_{3b} , are smaller than others. This result is favorable for the C_2 coexisting phase.

“ C_2 coexisting state” appears in the phase diagram:

Figure S10 a in the **SI E-2** exhibits the phase diagram derived from the minimization of the GL free energy exactly numerically. The horizontal (vertical) axis is the 2nd order GL parameter for the BO, a_1 (the cLC, a_2). Here, we use the GL parameters given in **Fig S9 a** and **b** at $T=0.01$. We find that the C_2 coexisting state is realized in the purple area. This C_2 coexisting phase appear when $T_{\text{BO}} > T_{\text{cLC}}$. **Figure S10 b** shows the obtained order parameters along the red broken line in **Fig. S10 a**. This C_2 coexisting state corresponds to the region $v > v^*$ in **Fig. 4 f** in the main text.

Why coexisting state for $T_{\text{cLC}} < T_{\text{BO}}$ becomes C_2 ?

In the **previous SI D**, we have already proved that the BO+cLC coexisting state becomes nematic (C_2 symmetry) when $T_{\text{cLC}} < T_{\text{BO}}$ owing to the 3rd order GL terms. In this proof, we assume the relation $b_1 \cdot b_2 < 0$. This relation is now numerically verified in the revised **Fig. S9 a**. To explain the reason for the C_2 coexisting phase more plainly, we added the following explanations on page 9:

“To understand the cLC+BO coexisting states in Fig. 4 f, the Ginzburg-Landau (GL) free energy analysis is very useful [10, 29, 30]. ... ”

In the revised **SI E-1**, we find that the free energy of the C_2 coexisting state is further lowered when the current order is $\boldsymbol{\eta} = (\eta, -\eta, 0)$ or $(\eta, \eta, -2\eta)$ when $T_{\text{cLC}} < T_{\text{BO}}$. In contrast, C_6 coexisting state appears when $T_{\text{cLC}} > T_{\text{BO}}$.

Change in the manuscript:

We explained the abovementioned results of the GL analysis in Section **E-1** and **E-2** in detail. Based on the present GL analysis and the DW equation analysis with self-energy correction, we revised the schematic phase diagram in **Fig. 4 f**.

○ Comments 1-7:

5. The relation between the current manuscript and Ref. 13 by the same authors is not fully clear to me. If I understand correctly, in Ref. 13 the authors demonstrate, by using a different kernel in Eq. (5), that (real) bond order forms in a system with an electronic structure very similar to the one studied in the current manuscript. In the current manuscript, it is the fluctuations of this bond order which are supposed to drive the formation of a loop-current order. However, with a different interaction kernel, what is the motivation for focusing on the bond-order fluctuations in the present manuscript? To demonstrate that the bond-order fluctuations are driving the formation of loop-current order in the first place, I think one should start from a scenario in which bond-order fluctuations are the strongest, as the one in Ref. 13. That being said, considering that the two interaction kernels are – at least to me – dissimilar, they produce very similar results for the real bond order [Fig. 2(c) of Ref. 13] and the imaginary bond order [Fig. 2(c) in

the present manuscript]. On the other hand, the real bond-order described in Fig. 2(c) of Ref. 13 is not the one assumed present in the current manuscript [Fig. 1(d)]. What is the reason for the similarity between the real bond-order of Ref. 13 and the imaginary bond-order in the current manuscript? Moreover, the authors should make clear the reasoning why bond-order fluctuations are expected to be dominant, when the interactions are different from those stated in Ref. 13.

○ Reply 1-7:

As we clearly explained in **Reply 1-4**, the BO fluctuations strongly develop in the HF approximation, the DW equation analysis, and the parquet RG analysis presented in **SI F**. In reality, the abundant BO fluctuations are reported by X-ray [48] and thermal transport [49] measurements in kagome metals. We find that the BO fluctuations induces the cLC order owing to the beyond-RPA effect. Below, we explain “*why odd-parity cLC order ($\delta t_{ij}^c = -\delta t_{ij}^e$) originates from the even-parity BO ($\delta t_{ij}^b = \delta t_{ij}^b$) fluctuations.*”

Why odd-parity cLC order is mediated by BO fluctuations?

The first term in **Fig. 4 a**, which represents the first-order MT term due to the BO fluctuations, gives the “repulsive Umklapp scattering”. Other odd-order MT terms cause similar contribution. The Umklapp scattering changes $f_{q\beta}^{AC}(\mathbf{k}_C)$ to $f_{q\beta}^{CA}(\mathbf{k}_A)$ in the DW equation. In contrast, the second term of **Fig. 4 a** and other even-order MT terms give the “attractive backward scattering”. Both scattering processes cooperate induce the odd-parity form factor $f_{q\beta}^{AC}(\mathbf{k}_C) = -f_{q\beta}^{CA}(\mathbf{k}_A)$. (It is noteworthy that the present cLC mechanism is reminiscent of the spin-fluctuation-induced cLC order in frustrated quasi-one-dimensional (1D) Hubbard model discussed in Ref.[66])

In **Fig.S4 b**, when $v \sim 1$, both BO and cLC susceptibilities cooperatively develop. As we explained in **Reply 1-4**, this relation is satisfied when the “attractive backward scattering” much larger than the “Umklapp scattering”, i.e., “(backward) \gg |(Umklapp)|”. This relation has been verified by the parquet RG analysis in the revised **SI F**.

The use of simple BO form factor g:

We have verified that the obtained cLC eigenvalue and form factor are not sensitive to

the long-range components of \mathbf{g} . Therefore, in the revised manuscript, we use a simple BO form factor \mathbf{g} composed of only the nearest bonds in the BO interaction in **Eq. (2)**. (Note that the long-range components of \mathbf{g} given by the DW equation depends on the model parameters. This is the reason why “the real bond-order described in Fig. 2(c) of Ref. 13 is not the one assumed present in the current manuscript [Fig. 1(d)].”)

Change in the manuscript:

We added the following explanations in the revised manuscript:

On page 7-8:

“Here, we discuss why the cLC order is mediated by the BO fluctuations. Let us consider the infinite series of MT terms in Fig.4 **a**, which is equal to $f_{\mathbf{q}_3}^{\text{AC}}((\lambda_{\mathbf{q}_3})^{-1}-1)$ according to the DW equation (7). The first term together with other odd-order MT terms give the “repulsive Umklapp” interaction, $\Gamma_{\text{um}}^{\text{MT}} > 0$, which leads to the odd-parity order $f_{\mathbf{q}_3}^{\text{AC}} = -f_{\mathbf{q}_3}^{\text{CA}}$. In contrast, the second term together with other even-order MT terms give the “attractive backward” interaction, $\Gamma_{\text{back}}^{\text{MT}} < 0$, which gives the attraction among the same $f_{\mathbf{q}_3}^{\text{lm}}$. Therefore, all series of MT terms cooperatively induce the odd-parity current order form factor shown in Fig. 3 **c**.”

On page 10-11:

“To verify the idea of the “BO fluctuation mediated cLC”, we perform the analysis of the parquet renormalization group (RG) formulation [10,71] and present the results in the revised SI F. A great merit of the RG method is that both particle-particle and particle-hole channels are treated on the same footing. We find that both BO and cLC fluctuations cooperatively develop in Fig. S11. This result of the RG study strongly supports the validity of the DW equation analysis.”

○ Comments 1-8:

6. The authors employ a temperature which is 10% of the hopping integral t (as stated in the section “BO form factor and fluctuations”). This is quite far from the low-temperature regime in which these orders appear in the experiments, and I would expect to see significant smearing from the use of such a high temperature. Indeed, in the units

provided, it corresponds to roughly 500 K (assuming 1 meV is 10 K which is a slight overestimate). The high temperature becomes entirely inappropriate when the authors claim that their results are consistent with the nematic order observed to onset near 35 K in CsV₃Sb₅ (in the section Z₃-nematic state given by the cLC-BO coexistence). The calculations need to be carried out at a much lower temperature if they are to be relevant for the experimental observations. While I am aware that RPA frequently overestimates transition temperatures substantially, I am not convinced that finite-temperature smearing has not had a significant impact on the results in this case.

○ Reply 1-8:

Energy-scale problem has been improved due to the self-energy:

Thank you very much for the constructive comment of the energy scale of the numerical study. To solve this problem, in the revised manuscript, we calculate the self-energy and the BO susceptibility self-consistently. Then, we introduce the obtained self-energy into the DW equation. The obtained transition temperatures is reduced to ~ 0.01 eV ≈ 100 K. Therefore, thanks to introducing the self-energy, the energy scale of the numerical results has been drastically improved. This is one of the main improvements achieved in the revised manuscript.

○ Comments 1-9:

7. The schematic phase diagram in Fig. 3(d) appears with no explanation of how the authors obtain it. Specifically, it is unclear from the manuscript how one goes from Fig. 2(b) and Fig. 3(c) to the schematic phase diagram. If I'm not mistaken, the calculations presented in the manuscript [specifically in Fig. 2(b)] are carried out in the disordered phase, i.e., the real bond order has not yet condensed (at least there is no mention of a Fermi surface reconstruction associated with such a condensation). The prominent bond-ordered dome in Fig. 3(d) leads the reader to believe otherwise. Furthermore, from Fig. 3(c) the leading eigenvalue is a strongly increasing function of v , in contrast to the dome-shape depicted in Fig. 3(d). Finally, as pointed out in Ref. 12, 3Q imaginary charge order (loop-current order) of the type studied here, cannot exist in the absence of a 3Q real charge order (bond order), as demonstrated by the presence of trilinear terms in Eq. (14)

in Ref. 12 and stated explicitly in the first column of p. 10 of Ref. 12. The authors have the same trilinear terms in their free energy in Eq. (S20). As a result, the right-hand side of the schematic phase diagram cannot be realized for the orders studied.

○ Reply 1-9:

The schematic phase diagram Fig. 4 f (=Fig. 3 d in the previous manuscript)

Thank you very much for the useful comment on the phase diagram.

We have revised the schematic phase diagram in **Fig. 4 f** in the revised manuscript, based on the numerical results of the DW equation analysis with the self-energy correction (**Figs. 4 d and e**) and the GL free energy analysis in **Fig. S10 a**. In the region $1/v < 1/v^*$ in **Fig. 4 f**, the BO+cLC coexisting state is nematic (C_2). In the opposite region $1/v > 1/v^*$, the coexisting state is C_6 .

In **Figs. 4 d and e** of the revised manuscript, we calculate the transition temperature T_{BO} and T_{cLC} . We discover that T_{cLC} tends to be higher than T_{BO} in the weak-coupling region ($v < v^*$). In contrast, the opposite relation is realized in the strong-coupling region ($v > v^*$) because the self-energy suppresses the cLC more strongly. The obtained results is schematically summarized in **Fig. 4 f**. Here, T_{BO} is defined as $\alpha_{BO} = \alpha_{BO}^*$ with $\alpha_{BO}^* = 0.985$ by considering the small inter-layer BO coupling $|v_{\perp}|$ ($\ll v$). The 3D BO is established when $\chi_g^{3D} = \chi_g^{2D}/(1 - \chi_g^{2D}|v_{\perp}|) = \infty$, that is, $|v_{\perp}| \sim (1 - \alpha_{BO})v$. Similar method is frequently used in deriving reliable T_{SDW} in spin fluctuation theories [67].

Interestingly, the obtained cLC+BO phase diagram is reminiscent of a typical SC-SDW phase diagram of spin-fluctuation-mediated superconductors, which is theoretically obtained by calculating the self-energy self-consistently [67].

To summarize, we succeeded in calculating the cLC and BO transition temperatures with including the self-energy effect. The obtained results are summarized in a schematic cLC+BO phase diagram in **Fig. 4 f**, with the aid of the GL analysis in the **SI E-1 and E-2**. This is a great progress achieved in the revised manuscript.

○ Comments 1-10:

Minor questions/comments:

m1. The kinetic Hamiltonian is never specified explicitly. I assume the authors are employing the standard one from, e.g., Eq. (2) of Phys. Rev. B 80, 113102 and that the filling factor quoted by the authors refers to total filling (spin up + spin down) and not filling per spin? Please clarify this. I think the authors should list the Hamiltonian in the methods section, so it is accessible to readers not familiar with the kagome lattice.

○ Reply 1-10:

Explanation of H_0 :

Thank you very much for this important comment.

We cite Phys. Rev. B 80, 113102 in introducing the kinetic term of kagome lattice model. We also introduce additional intra-sublattice hopping t' shown in **Fig.1 a** to reproduce the shape of the Fermi surface. The Fermi surface touch the vHS points when $n=n_{\text{vHS}}=0.917$, where n is the electron number per V site for both up and down spins. We added these explanations in the revised manuscript.

Change in the manuscript:

On page 3-4:

“... $h_{lm}^0(\mathbf{k})$ ($=h_{ml}^0(\mathbf{k})^*$) is the Fourier transform of the nearest-neighbor hopping integral t in Ref. [51] ...”

○ Comments 1-11:

m2. In the introduction, the authors motivate their reasons for considering the emergence of a charge-current loop phase within the bond-ordered phase. However, much of the motivation is only valid for the CsV3Sb5 compound. Both the K- and Rb-variants exhibit time-reversal symmetry-breaking at the charge-order transition (see Refs. 16, 17, and arxiv:2202.07713). Even in CsV3Sb5, the temperature at which time-reversal symmetry is broken is controversial. Some references report it to be at the charge-order transition (e.g., Refs. 17 and Phys. Rev. Research 4, 02324) while others (e.g., Refs. 14 and 15) report it to be below the charge-order transition. These subtleties are important for the reader to know. It should also be reflected in the introduction that this is a rapidly

progressing field, and many findings remain controversial. For instance, the observation of chiral charge order in Ref. 6 could not be reproduced in Ref 7.

○ Reply 1-11:

Explanation for experimental discrepancies:

Thank you very much for useful comments on experiments. We added several important experiments that were dropped in the previous manuscript.

Change in the manuscript:

On page 2:

“It has been reported by mu-SR study [14-17], Kerr rotation analyses [18], field-tuned chiral transport study [19], and STM study under magnetic field [4,19]. The transition temperature T_{TRSB} is close to T_{BO} by many experiments, while the TRSB order parameter is strongly magnified at $T^* \approx 35\text{K}$ for $A=\text{Cs}$ [15,17,19] and $T^* \approx 50\text{K}$ for $A=\text{Rb}$ [14]. Recently, magnetic torque measurement reveals the TRSB order associated with the rotational symmetry breaking, which is called the nematic order, at $T^* \approx 130\text{K}$ [20]. In contrast, TRSB was not reported by different experimental groups using the Kerr rotation [21] and STM [22] studies. Thus, the TRSB onset temperature is still under debate.”

Theory of the field-induced cLC order

The origin of such discrepancies between different experimental techniques attracts great attention now. Some of the measurements are performed under (tiny) magnetic field h_z . In trying to understand this problem, we recently discovered the “field-induced free-energy ΔF ” in arXiv: 2303.00623. The drastic impact of ΔF on the electronic states in kagome metals has been discovered. Even when $T_{\text{cLC}} < T_{\text{BO}}$ at $h_z = 0$, T_{cLC} increases to T_{BO} thanks to the cLC-BO- h_z trilinear coupling term in ΔF .

This drastic field-induced cLC order is realized when the cLC instability exists at $h_z=0$. Thus, the present BO fluctuation mediated cLC mechanism is important to understand the phase diagram under the magnetic field.

Change in the manuscript:

On page 11:

“Finally, we have recently revealed that the cLC order is strongly magnified under the magnetic field h_z [72]. Especially, in the case of $T_{BO} > T_{cLC}$ without h_z in Fig. 4 f, the cLC transition temperature increases to T_{BO} under $h_z \sim 1$ Tesla [72]. This analysis is consistent with the field-induced enhancement of the cLC order is observed by μ -SR studies [14,16,17] and field-tuned chiral transport study [19].”

○ Comments 1-12:

m3. The authors should specify in the text (not in the supplementary) what the \mathbf{a} -vector and the \mathbf{q}_n -vectors are. This is known by experts but not by all readers.

○ Reply 1-12:

Thank you very much for this kind comment. We revised the manuscript as follows:

Change in the manuscript:

We introduce the vector \mathbf{a}_{lm} and the vector \mathbf{q}_n in the caption of Fig. 1:

“ $2\mathbf{a}_{lm}$ is the unit cell vector, and we set $|2\mathbf{a}_{lm}|=1$. The relation $\mathbf{a}_{AB} + \mathbf{a}_{BC} + \mathbf{a}_{CA} = \mathbf{0}$ holds.”

“In kagome metals, \mathbf{q}_1 connects vHS-A and vHS-B and satisfied the relation $\mathbf{q}_1 = (2\pi/\sqrt{3})\mathbf{e}_z \times (2\mathbf{a}_{AB})$, where \mathbf{e}_z is the unit vector along \mathbf{z} -axis.”

○ Comments 1-13:

m4. The authors should specify what is meant by B_{3g} orbitals in this case. While experts will know that this refers to the irrep of D_{2h} , the site-symmetry group of the vanadium site, it is unlikely that most readers will understand this. Moreover, the authors should explain what relevance the orbital character of their one-orbital model has.

○ Reply 1-13:

In the present model, the main (pure-type) Fermi surface is composed of a single d-orbital without orbital degeneracy due to the crystalline electric field. Its orbital character (in this case, b_{3g} orbital) is not relevant for the theoretical results. Therefore, we simply explain that the present Fermi surface is composed of the single 3d-orbital model.

Change in the manuscript:

In the revised manuscript, we added the following explanation:

On page 3:

“(The d -orbital belongs to b_{3g} of D_{2h} point group, while its representation is not essential here.)”

On page 4:

“The FS around the van-Hove singular (vHS) point ($k = k_A, k_B,$ or k_C) is composed of a single 3d-orbital on V ion.”

○ Comments 1-14:

m5. While experts will be aware of the Ginzburg-Landau theory referenced in the section “BO fluctuation-mediated cLC order” this might not be the case for all readers. Hence, for the reader to properly understand the meaning of K3/2K4, the authors should include the expression for the free energy involving these coefficients. Moreover, details of the calculation of K3 and K4 should be included. Does this calculation depend on the bond-order fluctuations and Maki-Thompson diagrams?

○ Reply 1-14:

In the revised SI E-2, we added the expression of the GL free energy coefficients. The 3rd and 4th order coefficients are composed of the Green functions and the form factors g and f . On the other hand, the interaction I in the DW equation is not included. Therefore, the GL coefficients depends on the BO/cLC mechanisms indirectly, just through the form factors g and f .

Change in the manuscript:

In the revised SI E-2, we added the following explanation slightly below Eq. (S40):

“Note that these GL coefficients depend on the Green function (=bandstructure) and the form factors g and f . Thus, the GL coefficients depends on the BO/cLC mechanisms indirectly, just through the form factors.”

○ Comments 1-15:

m6. The numerical model, introduced in Supplementary Material A, requires more explanation. The authors should demonstrate that this model does not violate the symmetries imposed by the kagome lattice, specifically, it should respect six-fold symmetry in the disordered phase. The authors should clarify the conditions imposed on the momentum-space vectors by this fact.

○ Reply 1-15:

Thank you very much for this nontrivial question.

The answer is the following: The D_{6h} point group symmetry of original kagome lattice is not harmed in the present numerical study (such as the DW equation and the susceptibility) with using this square-lattice model. For example, the peak values of the DW equation eigenvalue $\lambda_{\mathbf{q}}$ at $\mathbf{q}=\mathbf{q}_1, \mathbf{q}_2, \mathbf{q}_3$ are completely equal (within the error $\sim 10^{-8}$). It is also the case for the BO susceptibility $\chi_g^0(\mathbf{q})$. Thus, we added the following explanation in the **SI A**, the first paragraph:

“We stress that the D_{6h} point group symmetry of original kagome lattice is not harmed in the present DW equation solution with using this square-lattice model.”

We present more detailed explanation for the validity of the DW equation analysis based on the square-lattice model in the “**SI of Author Reply C**” (on page 43).

Note that the velocity operator $v_x=dh_k/dk_x$, should be calculated based on the original kagome lattice structure. Therefore, numerical study of the anomalous Hall conductivity in **Fig. 6** were performed based on the original kagome lattice structure shown in **Fig 1**.

○ Comments 1-16:

m7. The authors stress that the method employed goes beyond RPA (i.e., beyond mean-field theory). From the derivation of Eq. (S15) it was not clear to me how this differs from RPA with a long-ranged interaction $V(\mathbf{q})$. The contribution from the Maki-Thompson diagrams is stated in Eq. (S12) with no additional explanation for the reader to understand the origin. Moreover, Fig. S2(b) represents the eigenvalues of an equation

that is never stated but, if I correctly, these are the solutions to the eigenvalue equation of Eq. (5) but with the interaction restricted to only Hartree-Fock terms? I think it would be very helpful to the reader if the authors made a similar plot for the interaction they employ in the main paper and include further details of the interaction kernel leading to Fig. S2(b). If the authors wish to make the claim that the method goes beyond RPA, it should be clear where these beyond-RPA contributions originate from, how the expressions are modified, and how the results are impacted.

○ Reply 1-16:

Difference between the present theory and RPA:

We would like to reply clearly to the question, “If the authors wish to make the claim that the method goes beyond RPA, it should be clear where these beyond-RPA contributions originate from, how the expressions are modified, and how the results are impacted”. We present the diagrammatic expression of “beyond RPA processes” in the present theory in **Fig S4 e** of the revised manuscript. They are also given in **Figs. R3 (d)** in the **SI of Author Reply B** (on page 42).

We also reply clearly to the important question, “From the derivation of Eq. (S15) it was not clear to me how this differs from RPA with a long-ranged interaction $V(q)$.” In the revised **Fig. 2 c** and **Fig. S4**, we express the BO fluctuation-mediated interaction as the yellow wavy lines. It gives the RPA susceptibility $\chi_w^{\text{RPA}}(q)$ for form factor $w (=g \text{ or } f)$ when “independent two bubbles are connected to both sides”, as we explain in **Fig. R3 (b)** in the **SI of Author Reply B**. On the other hand, it gives the DW equation when “both sides are connected by G^*w^*G ”, as we show in **Fig. R3 (c)**. By solving this DW equations, all the diagrams for $\chi_w^{\text{tot}}(q) = \chi_w^{\text{RPA}}(q) + \Delta\chi_w(q)$ are generated, where $\chi_w^{\text{RPA}}(q)$ is the RPA term and $\Delta\chi_w(q)$ is the beyond-RPA term. The latter is shown in **Figs. R3 (d)** or **Fig. S4 e** in the SI. Thus, “the present method goes beyond RPA” because $\Delta\chi_w(q)$ is totally dropped in the RPA..

Change in the manuscript:

We added the following explanation in the **SI C**, the second last paragraph:

“Figure S4 e presents the beyond-RPA processes $\Delta\chi_w(q)$ in the present study. The total

susceptibility is $\chi_w^{\text{tot}}(q) = \chi_w^{\text{RPA}}(q) + \Delta\chi_w(q)$. All these diagrams are generated by solving the DW equation. For $w=g$ (=BO form factor), $\chi_g^{\text{RPA}}(q)$ is large and positive, while $\Delta\chi_g(q)$ takes negative values due to the MT terms. For $w=f$ (=cLC form factor), $\chi_f^{\text{RPA}}(q)$ is zero, while $\Delta\chi_f(q)$ takes positive values, which becomes significant when $\alpha_{\text{BO}} \sim 1$. Therefore, the cLC susceptibility $\chi_f^{\text{tot}}(q)$ develops as large as the BO susceptibility in the present theory.”

○ Comments 1-17:

m8. In Fig. S2, there are three labels in the figure yet four in the figure text. Is there a label missing on the bottom figure?]

○ Reply 1-17:

Thank you very much for pointing out a mistake in the caption of **Fig.S2**. We corrected the mistake in the revised SI.

○ Comments 1-18:

m9. I feel the list of references do not accurately reflect the rapid progress in both experimental and theoretical work in this field. On the experimental side, this should be partly clear from the number of times in the above questions or comments where I reference publications that are not part of the references in the manuscript. On the theoretical side, the authors cite 20 of their own theoretical papers, while citing 13 other theoretical works in total. Of these, only five are specific to the kagome metals. While this could simply be a consequence of a very selective citation mentality, this seems contradicted by the 20 references to their own work.

○ Reply 1-18:

We have added the following important theoretical papers on kagome metals in the references and added the explanation in Introduction section.

- Mean-field theory of the cLC order: [28]
- Theories of BO+cLC coexisting states: [29,30]
- Theories of the pair-density-wave order: [45,46,47]

These papers were posted to arXiv after the first submission of our paper on 17 Jul, 2022 (arXiv:2207.08068) except for Ref. [45].

In addition, we dropped our previous works on the spin Hall effect that were included in the previous references.

○ Comments 1-19:

In summary, I cannot recommend publication of the manuscript at the current stage. As is evident from the questions and comments above, there are potentially severe issues that need to be addressed before the manuscript can be reconsidered. Questions 1-6 of the in the “Major questions/comments” have the potential to impact the validity of the conclusions and it is crucial that the authors address these. Moreover, as also explained above, the manuscript is hard to follow, and the connection between figures and the results presented in the text is not always apparent.

○ Reply 1-19:

Main achievements in the revised manuscript:

We are confident that we replied to the Major and Minor questions/comments satisfactorily. In addition, we have radically improved the manuscript to make the explanations much more understandable, as we explained in our reply. In addition, we have drastically improved our theoretical study to validate the idea of “BO fluctuation mediated cLC mechanism” by performing the following additional essential calculations.

- 【1】 Improvement of the main results (Figs. 3 and 4) by introducing the self-energy.**
- 【2】 Derivation of the phase diagram based on the GL free-energy analysis (Fig. S10).**
- 【3】 Verification of the main results by performing the parquet-RG analysis (Fig. S11).**

Thanks to the improvement **【1】**, we succeeded in obtaining reliable T_{BO} and T_{cLC} as function of the interaction v , as we shown in **Fig. 4 d** and **e** of the revised manuscript and in **Fig.S4** of the revised SI. Thanks to the improvement **【2】**, the existence of the “nematic BO+cLC coexisting state” is proved by the GL theory. Thus, we obtained the

convincing rationale of the schematic phase diagram in **Fig. 4 f**.

Thanks to **【3】**, we obtained the compelling evidence for the main message -- the “BO fluctuation mediated cLC mechanism” -- based on a different reliable theoretical framework. More quantitative results (such as the long-range component of the form factor) are derived from the DW equation analysis.

Since the main findings of the present theory have been proved in an unquestionable way and all the requirements by Reviewers #1 - #3 have been positively resolved, we are confident that the revised manuscript is appropriate for the publication in Nature Communications.

Change in the manuscript:

On page 6:

“To understand the BO+cLC phase diagram and the energy scale of these orders accurately, we have to include the self-energy that describes the quasiparticle properties. We calculate the on-site self-energy due to BO fluctuations (see Eq. (9) in Methods). The fluctuation-induced self-energy is essential to reproduce the T-dependence of various physical quantities, as well-known in spin fluctuation theories [63-65]. Here, we calculate $\chi_s(q)$ in Eq. (6) and $\Sigma_m(\epsilon_n)$ in Eq. (9) self-consistently.”

Reply to Reviewer #2

○ Comments 2-1:

The manuscript by Hiroshi Kontani et al proposes a mechanism for charge loop current (cLC) order due to bond order(BO) fluctuations, and discusses the consequence of the coexisting phase to understand the nematicity and anomalous Hall effect. It is motivated by the recently extensively studied Kagome metals compounds AV_3Sb_5 ($A=K, Cs, Rb$), which show a cascade of quantum phases.

○ Reply 2-1:

We are grateful to Reviewer #2 for reading our previous manuscript carefully and providing useful and constructive comments. We have improved the theoretical calculation intensively, by including the self-energy into our theoretical framework. By this improvement, the schematic phase diagram in **Fig. 4 f** is satisfactorily reproduced. In addition, the validity of the present theory has been confirmed by the RG analysis in the **SI F**. Owing to these improvements, we are confident that the revised manuscript becomes much more convincing.

○ Comments 2-2:

While the mechanism proposed in this work sounds interesting, I think the manuscript is not clear to follow (for one working in this field), and I do not fully understand and be convinced with the mechanism proposed in this manuscript based on the calculations presented. Therefore I cannot recommend the current version of the draft to publish in Nature Communications.

○ Reply 2-2:

Thank you very much for the comment “the mechanism proposed in this work sounds interesting”. We apologize that the explanations in the previous manuscript were not readable except for the specialists working on kagome metals. By referring to the useful comments by Reviewer #2, we have drastically improved the quality of the manuscript.

○ Comments 2-3:

First, the notations of the manuscript should be clearer to follow. For example, the terms “linearized DW equation”, “MT process”, “K3 & K4” etc used in the text are not defined. There also seems to be some subfigure missed in Fig. S2. The author also mentioned a few times the “paramagnon interference mechanism”, I think it should be explained for general readers in the field of correlated electron systems.

○ Reply 2-3:

We are grateful to Reviewer #2 for his/her constructive comments.

We have added the explanations of the following key concepts **(i)-(iv)** to make the manuscript considerably readable for general readers.

(i) Density-wave (DW) equation:

As we explained in “Universality of DW equation method” of **Reply 1-2**, the “DW equation” provides the general and useful formalism for general quantum phase transitions in metals, similarly to the Eliashberg gap equation for superconductivity [44]. The phase transitions in metals, not only CDW and SDW but also bond and current orders, are described as the symmetry breaking of the normal self-energy: $\Delta\Sigma \equiv \Sigma - \Sigma_{A1g}$ [44,50]. The DW equation enables us to derive the solution that satisfies the stationary condition of the free energy; $\delta F[\Delta\Sigma]/\delta(\Delta\Sigma) = 0$. To explain this fact, we have added the following explanations:

On page 3:

“The phase transitions in metals are described as the symmetry breaking of the normal self-energy; $\Delta\Sigma \equiv \Sigma - \Sigma_{A1g}$ [44,50]. $\Delta\Sigma$ is determined by the stationary condition of the free energy; $\delta F[\Delta\Sigma]/\delta(\Delta\Sigma) = 0$. The DW equations enables us to derive the solution that satisfies the stationary condition, as we proved based on the Luttinger-Ward theory [50]. Based on the DW equation, we discover that the odd-parity and TRSB $\Delta\Sigma$ is driven by the BO fluctuation exchange processes. (Note that the DW equation for $\Delta\Sigma$ is analogous to the Eliashberg equation for the superconducting (SC) gap Δ .)”

On page 7:

“By solving the DW equation (7), the optimized order parameter function is given as the eigenfunction $f_q^L(k)$ for the maximum eigenvalue λ_q . $\max_q \lambda_q = 1$ at the phase transition temperature. Note that $f_q^L(k)$ represents the symmetry breaking part in the normal self-energy $\Delta\Sigma(k, q) \sim \langle c_{k+q\sigma}^\dagger c_{k\sigma} \rangle$, and DW equation is directly derived from the stationary condition $\delta F[\Delta\Sigma]/\delta(\Delta\Sigma) = 0$ [50]. We can regard the DW equation (7) as the gap equation for the optimized p - h condensation [44,50].”

(ii) The Maki-Thompson (MT) term:

The MT term describes the scattering of electrons by bosonic fluctuations. This scattering process is important in metals near the quantum critical points. The MT term will be important in kagome metals due to the abundant BO fluctuations. We have added the following explanations on the MT term:

On page 6:

“The MT term describes the scattering of electrons due to the developed bosonic fluctuations. This scattering process is important in metals near the quantum critical points. For example, in nearly antiferromagnetic metals, the d-wave SC transition is induced by the MT processes of spin fluctuations. In kagome metals, the MT term represents the strong inter vHS scattering of electrons mediated by the abundant BO fluctuations; see Fig. 3 a. (The MT term also describes the s -wave SC state in kagome metals [11].) Here, we find that both $\tilde{\chi}_g$ and $\tilde{\chi}_{\text{cLC}}$ are comparably enlarged due to the MT processes in the present theory.”

On page 6-7:

“Thus, the cLC order should be ascribed to the beyond-HF mechanism. Here, we explain that the strong electron scattering between different vHS points due to the BO fluctuations, which are described as the MT processes, causes the odd-parity cLC order $\delta t_{ij}^s = -\delta t_{ji}^s$. (Note that the spin fluctuation exchange processes cause the cLC order in quasi-1D systems [66].)”

(iii) GL coefficients :

In the revised main text, we explain the 3rd order GL coefficients, b_1 and b_2 in detail, while the introduction of the 4th order terms (such as K3,K4) has been moved to the revised SI E-1.

(iv)paramagnon interference mechanism:

The “paramagnon interference mechanism” describes a “beyond RPA interaction” due to the second-order processes with respect to χ^s . This mechanism was developed to explain the nematic order in Fe-based SCs, and it was successfully applied to kagome lattice Hubbard model to explain the star-of-David BO. In the present manuscript, however, we introduced the effective BO interaction in **Eq. (2)**, instead of specifying the origin of the BO. Therefore, we present a minimum explanation for the paramagnon interference mechanism in the revised manuscript:

On page 5:

“In (i) [=paramagnon-interference due to on-site], Eq. (2) is induced by the spin-fluctuation-mediated beyond-RPA processes, the importance of which was originally revealed in the study of Fe-based SCs [44]. ”

○ Comments 2-4:

Second, the discussions on why the cLC order can be stabilized by the BO fluctuations should be expanded. It would be very helpful if more mathematical details can be provided about the role of 1st order BO and 2nd order BO fluctuations to stabilize the cLC. In fact, I have a hard time to understand how this mechanism is consistent with the g-ology study presented in Ref. 12 by Park et al. The 1st order term (Umklapp) correspond to the g3 in their Table I, 2nd order term correspond to the g2 in their Table I, it seems to me the claim made in the manuscript on page 6 and caption of Fig. 3 is not consistent with their observation (which is both g2 and g3 should be repulsive to stabilize the cLC order)?

○ Reply 2-4:

cLC order is obtained by the parquet RG method:

We are grateful to Reviewer #2 for suggesting us to consider the relation between the present theory and the renormalization group (RG) theory. To verify the idea of the “BO fluctuation mediated cLC”, we performed the analysis of the parquet RG formulation [10,71] and presented the results in the revised **SI F**. A great merit of the RG method is that both p-p and p-h channels are treated on the same footing. We find that both BO and cLC fluctuations cooperatively develop in **Fig. S11**. This result of the RG study strongly supports the validity of the DW equation analysis. This is one of the main achievements made in the revised manuscript.

In the revised **Fig. 4 a**, (= Fig. 3 a in the previous manuscript) we introduce the backward $\Gamma_{\text{back}}^{\text{MT}}$ and Umklapp $\Gamma_{\text{um}}^{\text{MT}}$ interactions. Here, $\Gamma_{\text{back}}^{\text{MT}}$ ($\Gamma_{\text{um}}^{\text{MT}}$) is composed of even (odd) order BO fluctuation terms. We find that the cLC instability $\Gamma_{\text{back}}^{\text{MT}} - \Gamma_{\text{um}}^{\text{MT}}$ takes a large value because $\Gamma_{\text{back}}^{\text{MT}} > 0$ and $\Gamma_{\text{um}}^{\text{MT}} < 0$.

In the RG scheme, $\Gamma_{\text{back}}^{\text{RG}} = -2g_1 + g_2$ and $\Gamma_{\text{um}}^{\text{RG}} = -g_3$ as we show in are show in **Fig. S11 b**. Therefore, the cLC instability is given as $G_{\text{cLC}}^{\text{RG}} = \Gamma_{\text{back}}^{\text{RG}} - \Gamma_{\text{um}}^{\text{RG}} = -2g_1 + g_2 + g_3$, which is consistent with Table I of Ref. [10] (=Ref. [12] in the previous manuscript). We find in **Fig. S11 c** that the renormalized interactions satisfy the relation $-g_1 \gg |g_2|, |g_3|$ when $g_1^0 = g_3^0 = -v/2$ (< 0), $g_2^0 = 0$, and $g_4^0 = U$ (> 0). Thus, $\Gamma_{\text{back}}^{\text{RG}} \gg |\Gamma_{\text{um}}^{\text{RG}}|$, and therefore $G_{\text{cLC}}^{\text{RG}}$ is strongly enlarged. As results, the cLC order is realized in both theoretical schemes.

Note that the Hartree (H) processes give $\Gamma_{\text{back}}^{\text{H}} = \Gamma_{\text{um}}^{\text{H}}$ ($\sim v/(1-v\chi_g^0) > 0$). Therefore, the relation $\Gamma_{\text{back}}^{\text{MT+H}} \gg |\Gamma_{\text{um}}^{\text{MT+H}}|$ is satisfied in the present theory. For this reason, the relation $T_{\text{cLC}} \sim T_{\text{BO}}$ is realized in **Fig. 4 d-e**.

Note that we made the replacements $I \rightarrow -I$ and $\{GG\} \rightarrow \{-GG\}$ in the prewise DW equations to make comparison with the RG theory. The DW equation is unchanged by these replacements. (Note that $T \Sigma_n \{-GG\} > 0$.)

Change in the manuscript:

In the revised manuscript, we added the following explanation why BO fluctuations

mediate the cLC order:

On page 7-8:

“Here, we discuss why the cLC order is mediated by the BO fluctuations. Let us consider the infinite series of MT terms in Fig.4 a, which is equal to $f_{q_3}^{AC}((\lambda_{q_3})^{-1}-1)$ according to the DW equation (7). The first term together with other odd-order MT terms give the “repulsive Umklapp” interaction, $\Gamma_{um}^{MT} > 0$, which leads to the odd-parity order $f_{q_3}^{AC} = -f_{q_3}^{CA}$. In contrast, the second term together with other even-order MT terms give the “attractive backward” interaction, $\Gamma_{back}^{MT} < 0$, which gives the attraction among the same $f_{q_3}^{lm}$. Therefore, all series of MT terms cooperatively induce the odd-parity current order form factor shown in Fig. 3 c.”

(It is noteworthy that the present cLC mechanism is reminiscent of the spin-fluctuation-induced cLC order in frustrated quasi-1D Hubbard model discussed in Ref.[66])

We also added the following explanation in terms of Γ_{back} and Γ_{um} :

On page 8:

“Figures 4 d and e indicate that both BO and cLC instabilities are comparable for $v^* \sim v$. Based on the parity argument, the BO (cLC) instability is given by $\Gamma_{back} + (-) \Gamma_{um}$. Therefore, the relation $\Gamma_{back} \gg |\Gamma_{um}|$ should be satisfied for $v^* \sim v$. In fact, the Hartree process gives positive $\Gamma_{back}^H = \Gamma_{um}^H \sim v/(1-v\chi_g^0)$, so the Hartree and MT processes strengthen each other in Γ_{back} but cancel each other in Γ_{um} . This relation is actually verified by the parquet RG study in the SI F”

○ Comments 2-5:

Third, to quantitatively support the key mechanism presented here, e.g. schematic phase diagram in Fig. 3d, I suggest the author to compute the transition temperature for the cLC near the quantum critical point of the BO, which can show if the dome for cLC really exist. Currently, from the results presented, I don't see what is the critical interaction v for the BO, and if the critical temperature is finite for the induced cLC at this critical v .

○ Reply 2-5:

The schematic phase diagram Fig. 4 f (=Fig. 3 d in the previous manuscript)

The revised **Figs. 4 d** and **e** show the calculated transition temperatures T_{BO} and T_{cLC} based on the DW equation with the self-energy correction. We discover that $T_{\text{cLC}} > T_{\text{BO}}$ in the weak-coupling region ($v < v^*$). In contrast, the opposite relation is realized in the strong-coupling region ($v > v^*$) because the self-energy suppresses the cLC more strongly.

We have revised the schematic phase diagram in **Fig. 4 f** in the revised manuscript, based on the numerical results of the DW equation analysis with the self-energy correction (**Figs. 4 d** and **e**) and the GL free energy analysis in **Fig. S10 a**. In the region $1/v < 1/v^*$ in **Fig. 4 f**, the BO+cLC coexisting state is nematic (C_2).

Main achievements in the revised manuscript:

We are grateful to this important suggestion by Reviewer #2.

After receiving the referee report, we have drastically improved our theoretical study to validate the idea of “BO fluctuation mediated cLC mechanism” by performing the following additional essential calculations:

- [1] Improvement of the main results (Figs. 3 and 4) by introducing the self-energy.**
- [2] Derivation of the phase diagram based on the GL free-energy analysis (Fig. S10).**
- [3] Verification of the main results by performing the parquet-RG analysis (Fig. S11).**

Thanks to the improvement **[1]**, we succeeded in obtaining reliable T_{BO} and T_{cLC} as function of the interaction v , as we shown in **Figs. 4 d** and **e** of the revised manuscript and in **Fig.S4** of the revised SI. Thanks to the improvement **[2]**, the existence of the “nematic BO+cLC coexisting state” is proved by the GL theory. Thus, we obtained the convincing rationale of the schematic phase diagram in **Fig. 4 f**.

Thanks to **[3]**, we obtained the compelling evidence for the main message -- the “BO fluctuation mediated cLC mechanism” -- based on a different reliable theoretical framework. More quantitative results (such as the long-range component of the form factor) are derived from the DW equation analysis.

Since the main findings of the present theory have been proved in an unquestionable way, we are confident that the revised manuscript is appropriate for publication in Nature Communications.

Reply to Reviewer #3

○ Comments 3-1:

Rina Tazai, Youichi Yamakawa, and Hiroshi Kontani report on the role of bond-order fluctuations for the charge-loop current order and Z3-nematicity in kagome metal AV₃Sb₅ (A = Cs,Rb,K). This is an interesting work addressing an important question about the interplay between charge-loop current and the bond-order. The paper will be of interest to both experimentalists and theorists. But before I can recommend the paper for publication, I have some comments/questions, shown below:

○ Reply 3-1:

We are grateful to Reviewer#3 for reading our previous manuscript carefully and for finding broad interest in our theoretical study. We have improved the manuscript intensively by adding important discussions and new figures. Especially, we found the importance of the BO fluctuation induced self-energy, by referring to the established spin fluctuation theories on superconductivity [63-65]. By including the self-energy, we succeeded in explaining the experimental P-T phase diagram, as we show in **Figs. 4 d-f** in the revised manuscript. In addition, the validity of the present theory has been confirmed by the RG analysis in the **SI F**. Owing to these improvements, we are confident that the revised manuscript becomes much more convincing.

○ Comments 3-2:

1. According to muon-spin rotation (μ SR) measurements the time-reversal symmetry breaking state seems to occur at the onset of charge order in all three compounds KV₃Sb₅ (Nature 602, 245-250 (2022)), RbV₃Sb₅ (<https://arxiv.org/abs/2202.07713>) and CsV₃Sb₅ (Phys. Rev. Research 4, 023244). This is also supported by Kerr effect measurements, as mentioned by the authors as well. It is indeed true that the magnetic response gets stronger at some lower temperature. For instance, in RbV₃Sb₅ very well pronounced two-step increase of the magnetic relaxation rate was reported by μ SR with two characteristic temperatures of $T1^* = 110$ K (onset temperature for charge order) and

$T_2^* \approx 50$ K (<https://arxiv.org/abs/2202.07713>). In the same work, authors followed the TRSB charge order transitions in RbV3Sb5 as a function of pressure. The obtained phase diagram shows that the two-step transition becomes a single time-reversal symmetry-breaking transition at 1.58 GPa, above which $T_1^* = T_2^*$ shows a faster suppression and follows the phase boundary of the charge order. Two transitions with $T_1^* = 90$ K and $T_2^* \approx 30$ K are also observed in CsV3Sb5. This shows that time-reversal symmetry breaking signal appears right at the onset of charge order and gets enhanced below 50 K in RbV3Sb5 and below 30 K in CsV3Sb5. How do these experimental observations fit into the picture of bond-order fluctuations mediated loop current? Authors should extend the discussion on this aspect and more clearly discuss the connection between the predicted phase diagram shown in Figure 3d and the experimental observations.

○ Reply 3-2:

The schematic phase diagram Fig. 4 f (=Fig. 3 d in the previous manuscript)

We have revised the schematic phase diagram in **Fig. 4 f** in the revised manuscript, based on the numerical results of the DW equation analysis with the self-energy correction (**Figs. 4 d** and **e**) and the GL free energy analysis in **Fig. S10 a**. In the region $1/\nu < 1/\nu^*$ in **Fig. 4 f**, the TRSB occurs in the nematic (C_2) BO+cLC coexisting state.

Experimental P-T phase diagram:

Thank you very much for the very important comments on the experimental phase diagram in kagome metals. The T_{cLC} obtained by the present theory corresponds to the characteristic temperature T_2^* ($< T_{\text{BO}}$) reported in Ref. [14]. In our theory, the TRSB occurs below T_{cLC} due to the cLC order. This fact naturally explains the prominent TRSB observed below T_2^* experimentally in Ref. [14].

To explain the consistency between the theory and the experiment, we show the newly obtained numerical results in **Figs. R1 (a)** and **(b)** in this Author Reply (= **Fig. 4 e- d** in the main text). The experimental P-T phase diagram with T_{BO} ($= T_1^*$) and T_{cLC} ($= T_2^*$) is shown in **Fig. R1 (c)** in Ref. [14]: It looks very similar to **Fig. R1 (b)** once the suppression of the cLC order below T_{BO} is taken into account.

We stress that the relation $T_{\text{BO}} > T_{\text{cLC}}$ in the strong correlation region $\nu > \nu^*$ is obtained by introducing the self-energy effect in the revised manuscript. This is the main progress

made in the revised manuscript.

Figure R1:

Change in the manuscript:

We added the following sentences to explain this progress in the revised manuscript:

On page 2:

“It has been reported by μ -SR study [14-17], Kerr rotation analyses [18], field-tuned chiral transport study [19], and STM study under magnetic field [4,19]. The transition temperature T_{TRSB} is close to T_{BO} by many experiments, while the TRSB order parameter is strongly magnified at $T^* \approx 35K$ for $A=Cs$ [15,17,19] and $T^* \approx 50K$ for $A=Rb$ [14]. Recently, magnetic torque measurement reveals the TRSB order associated with the rotational symmetry breaking, which is called the nematic order, at $T^* \approx 130K$ [20]. In contrast, TRSB was not reported by different experimental groups using the Kerr rotation [21] and STM [22] studies. Thus, the TRSB onset temperature is still under debate.”

On page 8-9:

“We discuss that Figs. 4 **d** and **e** naturally explain experimental P-T phase diagram with T_{BO} and T_2^* ($\sim T_{TRSB}$) given by μ SR study [14] for $A=Rb$, considering that v/W_{band} decreases with P. A schematic BO+cLC phase diagram derived from the present theory is depicted in Fig. 4 **f**. (This schematic phase diagram is supported by the Ginzburg-Landau (GL) analysis in Figs. S10 **a-c** in the SI E-2. The suppression of the secondary order due to the primary order is considered.) The cLC phase is realized next to the BO phase because it is mediated by the BO fluctuations. This cLC+BO phase diagram is reminiscent of the SC+SDW phase diagram of spin-fluctuation-mediated

superconductors, which has been reproduced by considering the self-energy.”

Theory of field-induced cLC order:

The origin of such discrepancies between different experimental techniques attracts great attention now. Some of the measurements are performed under (tiny) magnetic field h_z . To try to understand this problem, we recently studied the “field-induced free-energy ΔF ” in arXiv: 2303.00623. The drastic impact of ΔF on the electronic states in kagome metals has been discovered. Even when $T_{\text{cLC}} < T_{\text{BO}}$ at $h_z = 0$, T_{cLC} increases to T_{BO} thanks to the “cLC-BO- h_z trilinear coupling term” in the free-energy.

This drastic field-induced cLC order is realized when the cLC instability exists at $h_z = 0$. Thus, the present BO fluctuation mediated cLC mechanism is important to understand *the phase diagram under the magnetic field*.

Change in the manuscript:

On page 11:

“Finally, we have recently revealed that the cLC order is strongly magnified under the magnetic field h_z [72]. Especially, in the case of $T_{\text{BO}} > T_{\text{cLC}}$ without h_z in Fig. 4 f, the cLC transition temperature increases to T_{BO} under $h_z \sim 1$ Tesla [72]. This analysis is consistent with the field-induced enhancement of the cLC order observed by μ -SR studies [14,16,17] and field-tuned chiral transport study” [19]”

○ Comments 3-3:

2. From the experimental front, at ambient pressure a nodeless SC gap is reported for CsV3Sb5 and nodal SC pairing for RbV3Sb5 and KV3Sb5 (<https://arxiv.org/abs/2202.07713>). But once charge order is either strongly suppressed or fully suppressed, the nodeless state is stabilised for all three compounds. So, it seems that depending on the fine details of the charge ordered state in AV3Sb5, either nodal or nodeless state can be found. It would be helpful if authors can comment on the superconducting gap symmetry in these kagome materials from the perspective of bond-order fluctuations theory. Is there a parameter space of charge order in the theoretical model in which a nodal gap is stabilized?

○ Reply 3-3:

Explanation of the P-induced nodeless SC gap:

Thank you very much for this very important referee comment.

In the BO fluctuation theory published in Ref. [11], anisotropic s-wave SC gap is mediated by BO fluctuation exchange processes. The nodal \rightarrow nodeless crossover is realized by increasing the strength of the BO fluctuations or introducing the impurity scattering. The BO fluctuations strongly develop around the BO phase boundary ($T_{BO} \sim 0$) under the external pressure.

Figure R2 (a) show the eigenvalue of the SC gap equation obtained in Ref. [11] at a fixed temperature. When the strength of the BO fluctuations is moderate, the nodal s-wave singlet SC state (**Fig. R2 (b)**) is realized. With increasing the BO fluctuation strength, the s-wave T_{SC} increases and the gap structure changes to nodeless (**Fig. R2 (c)**). Thus, the present BO fluctuation theory naturally explain the observation in <https://arxiv.org/abs/2202.07713> for A=Rb and K. Thus, both the cLC order and the superconductivity in kagome metals are explained by the BO fluctuation theory on a same footing.

We stress that the impurity induced isotropic SC gap in A=Cs reported in Refs . [12] and [13] is also understood based on the BO fluctuation theory.

Figure R2:

Change in the manuscript:

To explain the experiments of the SC state in kagome metal, we added the following sentences in Introduction on page 2:

“Below T_{BO} , superconductivity with anisotropic gap emerges for $A=Cs$ [12,13], and the gap structure changes to isotropic by introducing impurities. Also, nodal to nodeless crossover is induced by the external pressure in $A=Rb,K$ [14]. These results are naturally understood based on the BO fluctuation mechanism [11].”

Main achievements in the revised manuscript:

After receiving the referee report, we have drastically improved our theoretical study to validate the idea of “BO fluctuation mediated cLC mechanism” by performing the following additional essential calculations:

- 【1】** Improvement of the main results (Figs. 3 and 4) by introducing the self-energy.
- 【2】** Derivation of the phase diagram based on the GL free-energy analysis (Fig. S10).
- 【3】** Verification of the main results by performing the parquet-RG analysis (Fig. S11).

Thanks to the improvement **【1】**, we succeeded in obtaining reliable T_{BO} and T_{cLC} as function of the interaction v , as we shown in **Figs. 4 d** and **e** of the revised manuscript and in **Fig.S4** of the revised SI. Thanks to the improvement **【2】**, the existence of the “nematic BO+cLC coexisting state” is proved by the GL theory. Thus, we obtained the convincing rationale of the schematic phase diagram in **Fig. 4 f**.

Thanks to **【3】**, we obtained the compelling evidence for the main message -- the “BO fluctuation mediated cLC mechanism” -- based on a different reliable theoretical framework. More quantitative results (such as the long-range component of the form factor) are derived from the DW equation analysis.

Since the main findings of the present theory have been proved in an unquestionable way, we are confident that the revised manuscript is appropriate for publication in Nature Communications.

List of main changes made in the revised manuscript:

The main changes made in the revised manuscript have been explained in the Reply to each Reviewer. The revised parts are shown in red characters in the revised manuscript. Other main changes are listed below:

○ Revised Figures:

- Figure 1: **c** The FS is changed to $(t, t') = (-0.5, -0.08)$.
- Figure 2: Expressions of **a** the BO fluctuations and **d** the self-energy are added.
- Figure 3: Numerical results in **b** and **c** are replaced with the “improved calculations with the self-energy correction”.
- Figure 4: **a** Detailed explanations are added. **b-e** Improved calculations with the self-energy correction are presented. **f** Schematic phase diagram is revised. We find that both cLC and BO fluctuations are strongly magnified for $v \sim v^*$.
- Figure 6: Revised anomalous Hall conductivities are presented.
- Figure S4: Results of the self-consistent DW equation, in which the renormalized BO susceptibility is calculated self-consistently. The obtained relation $T_{\text{cLC}} \approx T_{\text{BO}}$ is consistent with Figs. 4 **d-f** in the main text.
- Figure S9: GL coefficients derived from the diagrammatic method.
- Figure S10: The phase diagram obtained by minimizing the GL free energy numerically, as functions of the 2nd GL coefficients for the BO (a_1) and the cLC (a_2). The nematic BO+cLC coexisting phase is realized for $T_{\text{BO}}^0 \gtrsim T_{\text{cLC}}^0$.
- Figure S11: Analysis of the parquet RG equation for kagome metals. The strong increment of the renormalized G_{cLC} and G_{cBO} in **d** and **f** means that both cLC and BO fluctuations are magnified cooperatively.

○ Revised Supplemental Information:

SI C: Self-consistent DW equation method: derivation of renormalized BO fluctuations

SI E-1: Stability of the nematic BO+cLC state: Analytic discussion

SI E-2: Stability of the nematic BO+cLC state: Numerical study

SI F: Parquet RG theory for kagome metals: Derivation of cLC and BO instabilities

SI of Author Reply A: Derivation of scalar $\chi_g(q)$ based on the matrix expression:

Here, we prove the expression of the scalar BO susceptibility in Eq. (6), $\chi_g = \chi_g^0 / (1 - v \chi_g^0)$, based on the matrix expression. For simplicity, we introduce the following 2×2 expression of the irreducible susceptibility and the BO interaction in Eq. (2) at $q=q_1$:

$$\hat{\chi}_g^0 = \begin{pmatrix} \chi_g^{0,AB,AB} & \chi_g^{0,AB,BA} \\ \chi_g^{0,BA,AB} & \chi_g^{0,BA,BA} \end{pmatrix} \quad \begin{array}{c} l \quad \quad m \\ \bullet \text{---} \bullet \\ l' \quad \quad m' \end{array}$$

$$\hat{V} = v \begin{pmatrix} 1 & 1 \\ 1 & 1 \end{pmatrix} \quad \begin{array}{l} (l, l') = (A, B), (B, A) \\ (m, m') = (A, B), (B, A) \end{array}$$

Here, $\chi_g^{0,lr,mm'}$ is given in Eq. (5) in the main text, and the first and the second row correspond to (A,B) and (B,A), respectively. The BO interaction in Eq. (2) is independent of the spin index. By considering the factor 2 due to the spin index of each χ_g^0 , the 2×2 expression of the charge-channel susceptibility is given as

$$\hat{\chi}_g = \hat{\chi}_g^0 \cdot (\hat{1} - \hat{V} \cdot \hat{\chi}_g^0)^{-1}. \quad (\text{R1})$$

This susceptibility diverges when

$$\det(\hat{1} - \hat{V} \cdot \hat{\chi}_g^0) = 0. \quad (\text{R2})$$

From Eq. (R2), we obtain $1 - v\chi_g^0 = 0$, where $\chi_g^0 \equiv \chi_g^{0,AB,AB} + \chi_g^{0,AB,BA} + \chi_g^{0,BA,AB} + \chi_g^{0,BA,BA}$ corresponds to Eq. (4) in the main text. Thus, Eq. (R1) diverges when the BO Stoner factor $\alpha_{\text{BO}} = v\chi_g^0$ reaches unity. Therefore, the BO instability first diverges with increasing v .

From Eq. (R1), we can verify that $\chi_g \equiv \chi_g^{AB,AB} + \chi_g^{AB,BA} + \chi_g^{BA,AB} + \chi_g^{BA,BA}$ exactly reproduces Eq. (6); $\chi_g = \chi_g^0 / (1 - v \chi_g^0)$. Thus, Eq. (6) in the main text is proved in terms of the matrix expression.

SI of Author Reply B: MT-type interaction due to BO fluctuations exchange processes:

Figure R3:

(a) The BO fluctuations mediated interaction W in the RPA due to Hartree term. g is the BO form factor. The scalar irreducible susceptibility $\chi_g^0(q)$ is given in Eq. (4).

(b) RPA susceptibility for the form factor w , $\chi_w^{\text{RPA}}(q)$. When $w = \text{“onsite”}$ form factor, $\chi_{w,g}^0(q_m) \equiv -T \sum w(GG)g = 0$ and therefore $\chi_w^{\text{RPA}}(q_m) = \chi_w^0(q_m)$. Thus, only the BO susceptibility $\chi_{w=g}^{\text{RPA}}(q)$ is enlarged in the Hartree approximation. (Note find numerically that $\chi_{w,g}^0$ is zero or just $\chi_g^0/10 \sim \chi_g^0/100$ for $w = \text{“onsite”}$ form factor.)

(c) The DW equation with the MT kernel function. w is the eigenfunction. This expression is similar to the Eliashberg gap equation.

(d) The corrections for the $\chi_w^{\text{RPA}}(q)$ due to the beyond-RPA processes, $\Delta\chi_w(q)$. The total susceptibility is $\chi_w^{\text{tot}}(q) = \chi_w^{\text{RPA}}(q) + \Delta\chi_w(q)$. All these diagrams for $\chi_w^{\text{tot}}(q)$ are generated by solving the DW equation in (c).

For $w=f$ (=cLC form factor), $\chi_f^{\text{RPA}}=0$, while $\Delta\chi_f$ takes positive values. $\Delta\chi_f$ is significant when $\alpha_{\text{BO}} \sim 1$. Therefore, the cLC susceptibility $\chi_f^{\text{tot}}(q)$ develops as large as the BO susceptibility in the present theory.

SI of Author Reply C: Why D_{6h} symmetry holds in the “square kagome lattice model”:

In Fig. R4 (a), purple broken line represents the Brillouin zone (BZ) of the “square kagome lattice model”. Another equivalent BZ is shown by green line. In Fig. R4(b), green line represents the BZ of the “original hexagonal kagome lattice”, which is deformed to the green line BZ in (a) by the linear transformation U . The nesting vectors $\mathbf{q}_1=(\pi,0)$, $\mathbf{q}_2=(\pi, \pi)$, $\mathbf{q}_3=(0,\pi)$ in Fig R4(a) move to those in Fig. R4(b) by U . Figure R4 (c) shows the \mathbf{k} -points for the BZ in Fig. R4 (a). Figure R4 (d) shows the \mathbf{k} -points for the BZ in Fig. R4 (b).

In Fig. R4 (a), we superimpose the irreducible BO susceptibility $\chi_g^0(\mathbf{q})$ calculated by using the \mathbf{k} -points inside the BZ in Fig. R4 (c). The peak values of $\chi_g^0(\mathbf{q})$ at $\mathbf{q}=\mathbf{q}_n$ ($n=1,2,3$) are equivalent. In Fig. R4 (b), we also superimpose $\chi_g^0(\mathbf{q})$ calculated by using the \mathbf{k} -points inside the BZ in Fig. R4 (d). The obtained $\chi_g^0(\mathbf{q})$ at $\mathbf{q}=\mathbf{q}_n$ is equivalent to that in Fig. R4 (a). Thus, the D_{6h} symmetry of the original hexagonal kagome lattice is not harmed in the numerical study using the \mathbf{k} -points in Fig. R4 (c). In fact, the Green function $G_{lm}(\mathbf{k},\varepsilon)$ in the square kagome lattice model is equal to $G'_{lm}(\mathbf{k}',\varepsilon)$ in the original hexagonal kagome lattice model for $\mathbf{k}'=U\mathbf{k}$ in the case of $l=m$. The difference in the phase factors for $l\neq m$ are cancelled out in the physical quantities.

REVIEWER COMMENTS

Reviewer #1 (Remarks to the Author):

I appreciate that the authors took the time to consider all my questions in detail in their response. I think I now understand their approach better. However, I am unable to recommend publication at the present time. The main finding of the manuscript is the close relation between bond order and loop-current order and showing that loop-current order can be driven by bond-order fluctuations. However, this close relationship was already established early on by Refs. 10 and in PRB 104, 045122 (2021) and here it was also argued that bond order promotes loop current order. While the present manuscript goes beyond the patch models studied in those references, it does not add any new information and it does not provide a more material specific prediction, as it does not employ the full band structure of the AV3Sb5 materials. The interaction used in Eq. (2) essentially guarantees that the model will find either bond order or a coexistence phase with bond order and loop current order, like the cases studied in, e.g., Ref. 10. In terms of relevance beyond the AV3Sb5 kagome metals, I am not convinced by the authors' reply. The method itself is not novel and has been applied by the authors in many of their previous works, as evidenced by the list of references. While other metallic kagome compounds have been discovered (e.g., 166s and 11s), these appear further from the van Hove singularity and, to my knowledge, do not exhibit the same type of charge order. Consequently, I don't find a reason why the manuscript should be published in Nature Communications over a more specialized journal. Finally, while I appreciate the authors' efforts to make the manuscript more readable, I still find it very hard to follow. I think including some of the important technical details from the supplementary would help the reader understand the approach.

Below I provide responses to the authors answers in the cases where I believe there are still outstanding issues:

1-3: If I understand correctly, the authors argue that bond order is dominant in the 3-site kagome model near the upper van Hove singularity, and that this therefore provides a good starting point for the study of how bond-order fluctuations develop in the AV3Sb5 kagome metals. I have several problems with this argument. First, in the stated references [6,7,11], I am not sure exactly which parts the authors refer to. In [6,7], both fRG studies, the phenomenology is richer than only bond order, and there are multiple instabilities that the authors do not consider in the present manuscript. This fact should be made apparent in the text, bond order is not the only instability occurring near the pure van Hove singularity, it is very sensitive to interactions, as also shown in, e.g., Ref 10. In [11], if I understand correctly, the authors find that the naïve 3-site model does actually not support bond order unless additional interactions are included (coming from specific Maki-Thompson and Azlamazov-Larkin terms). From Ref. [11] it is not clear to me why one should expect such terms to play an important role in either the AV3Sb5 compounds or in the 3-site kagome

model. Furthermore, while I appreciate that the authors aim to study the impact of bond-order fluctuations, they are not ruling out that other effects could be responsible for the experimental observations. Indeed, as they start from Eq. (2), they only allow bond-order fluctuations to contribute and neglect the influence from any other channel. Allowing inter- and intra-sublattice channels to compete by including an additional Fermi surface would provide a more complete picture of the AV₃Sb₅ materials.

1-4: I understand the interaction in Eq. (2) ensures that only bond-order contributions enter the susceptibility since v only parameterizes this interaction. However, it is very hard for the reader to know what v corresponds to microscopically (i.e. in terms of e.g. Coulomb interactions). I think the paper would improve significantly if the authors would provide a relationship between v and electronic interactions in the main text. For instance, the interaction between the bond-order fluctuations in the patch model of Ref. 10 are ascribed to specific intra- and inter-patch interactions, which makes it very easy for the reader to identify the ingredients necessary to promote such bond-order fluctuations. Would it be possible for the authors to provide a similar relationship between interactions and their parameter v ? They hint at it below Eq. (2) but it is never made explicit. The associated supplementary B-2 is very opaque and hard to follow. In my opinion, the authors could improve the text by integrating part of that supplementary into the main text and make succinct statements about how v is related to the microscopic parameters both in the exact and approximate scenarios. This would strengthen the paper and make it easier for the reader to understand. While I appreciate γ is a bit harder to pin down, I think it would make the results more compelling if the authors explored a larger parameter regime in γ to demonstrate the robustness of their findings. For instance, by making a three-dimensional plot containing v , γ , and T (Figs. 4 (d) and (e) provides an indication but more values of γ is desirable).

1-6: I think my confusion in this question stemmed from the authors referring to the nematic state as a 3Q state which, in my mind implies that three order parameters have the same magnitude. I appreciate now that this is not what the authors had in mind. Regarding the schematic phase diagram, only the leading instability is obtained from the density-wave equation, the number of Q-vectors condensing is, as far as I can tell, determined from the Landau free energy. Since the number of possible phases is vast, the authors should provide the details of the calculations of the coefficients of the free energy which supports their assertions that the stabilized phases are indeed the ones appearing in the schematic phase diagram.

1-13: This is the site-symmetry group of the vanadium atoms (or more generally, the sites of the kagome lattice). D_{6h} is the point group.

Reviewer #2 (Remarks to the Author):

The revised manuscript by Rina Tazai et al have improved significantly to make their proposal of BO fluctuation induced cLC order more convincing. All my detailed questions and comments have been addressed carefully by the authors. Therefore, I would recommend the manuscript to be publish in Nature Communications.

While reading the revised manuscript, I noticed some of the notations remain undefined. While the readability of the manuscript has been improved, I believe the authors should make another effort to check the presentation of the work and make sure all quantities have been defined properly in the text (say a parameter γ in figure 4 (d, e)), and all the figures referred in the text are presented (say Fig. S2 d referred in the SM is not shown). Such effort would increase the impact of the work.

Reviewer #3 (Remarks to the Author):

The authors have discussed the criticism by reviewers thoroughly and very diligently. The revision improves the quality of the paper. I recommend this manuscript to be published in Nature Communication as it is.

Reply to Comments by Reviewer #1

○ Comment #1-1

I appreciate that the authors took the time to consider all my questions in detail in their response. I think I now understand their approach better. However, I am unable to recommend publication at the present time. The main finding of the manuscript is the close relation between bond order and loop-current order and showing that loop-current order can be driven by bond-order fluctuations. However, this close relationship was already established early on by Refs. 10 and in PRB 104, 045122 (2021) and here it was also argued that bond order promotes loop current order. While the present manuscript goes beyond the patch models studied in those references, it does not add any new information and it does not provide a more material specific prediction, as it does not employ the full band structure of the AV₃Sb₅ materials. The interaction used in Eq. (2) essentially guarantees that the model will find either bond order or a coexistence phase with bond order and loop current order, like the cases studied in, e.g., Ref. 10. In terms of relevance beyond the AV₃Sb₅ kagome metals, I am not convinced by the authors' reply. The method itself is not novel and has been applied by the authors in many of their previous works, as evidenced by the list of references. While other metallic kagome compounds have been discovered (e.g., 166s and 11s), these appear further from the van Hove singularity and, to my knowledge, do not exhibit the same type of charge order. Consequently, I don't find a reason why the manuscript should be published in Nature Communications over a more specialized journal. Finally, while I appreciate the authors' efforts to make the manuscript more readable, I still find it very hard to follow. I think including some of the important technical details from the supplementary would help the reader understand the approach.

○ Reply #1-1

We are grateful to Reviewer #1 for reviewing the previous two manuscripts very carefully. Both the previous and the present comments by Reviewer #1 have been very useful to improve the quality of the manuscript. Thank you very much for the comment “*I appreciate that the authors took the time to consider all my questions in detail in their response. I think I now understand their approach better.*”

The current most significant issue of AV₃Sb₅ is the “spontaneous charge loop current (cLC) order with time-reversal-symmetry-breaking (TRSB)”, while its driving force is totally unknown. To solve this hot issue, we provide a promising “novel idea: the bond-order (BO) fluctuation

mediated current order mechanism”. The present mechanism assumes the existence of the BO order and fluctuations that are robustly observed in kagome metals experimentally. The present mechanism is general because it is independent of the microscopic origin of the BO order/fluctuations. For this reason, the present theoretical idea will be useful to understand the current order in various metals. We revised the manuscript very eagerly to attract general readers in the wide field of condensed matter physics.

Hereafter, we would like to reply to the important comments and questions mentioned by Reviewer #1:

“Complemental relationship” between two reliable theoretical methods: the present theory and the RG study:

We consider that the Feynman diagram theory (used in the present study) and the RG theory are complementary based on our experience of the functional RG studies [37,38,45,67]. Comparative studies of both theories yield physically reliable findings. A great advantage of the RG study is that the all two-particle scatterings (both Cooper and Peierls channels) are calculated in an unbiased way while using several simplifications. Various leading instabilities are obtained by changing the initial interaction Hamiltonian (*e.g.*, $g_1^0, g_2^0, g_3^0, g_4^0$ in parquet RG). However, it is frequently nontrivial to pin down the fundamental “physical process” for the obtained results when the vertex corrections are essential.

In the present study, we discover that the current order instability is driven by the “BO fluctuations exchange process”: This is a physical process that gives the mathematical relation for the current order “ $-g_1 \gg g_2, g_3, g_4$ ”; see Fig. S11. The present method enables us to calculate reliable T_{BO} and T_{CLC} in Fig. 4 based on a realistic 3-site tight-binding model. The agreement between the two reliable theoretical methods strongly assures the reliability of the “BO fluctuation mediated cLC mechanism”. This novel concept was proposed for the first time in the present study, thanks to the combination with the RG study in Fig.S11.

To explain this consideration, we added the following paragraph on page 11.

”We comment on the complementary relationship between the present theory and the parquet RG theory. The latter theory solves a simplified 3-patch model in an unbiased way, leading to the development of both CLC order and BO, while the relationship between the two orders is not clear. On the other hand, the present theory focuses on the existence of the experimental BO phase and reveals that abundant BO quantum fluctuations lead to TRSB particle-hole condensation. Thus, the concept of the “BO fluctuation mediated cLC” has been verified

based on different reliable theories.”

Comments on other kagome metals:

Thank you very much for the comment on the recently discovered double-layer kagome metal ScV_6Sn_6 , which shows $\sqrt{3}\times\sqrt{3}$ CDW and the spontaneous TRSB state. As for $\sqrt{3}\times\sqrt{3}$ CDW order, the nesting between the vHS points at M points would be unimportant, as mentioned by Reviewer #1. The difference in the CDW wavevector would originate from the CDW mechanism. In fact, the impact of the “softening of a flat phonon mode for ScV_6Sn_6 ” was reported by experimental [78] and theoretical [77] papers. (Such a flat phonon mode is absent in AV_3Sb_5 .)

Because the present cLC mechanism is independent of the origin of BO fluctuations, the present idea is expected to be applicable to the cLC order observed in ScV_6Sn_6 .

To explain this consideration, we added the following paragraph on pages 11-12.

“Finally, we comment on some interesting kagome metals other than AV_3Sb_5 . Double-layer kagome metal ScV_6Sn_6 shows $\sqrt{3}\times\sqrt{3}$ charge-density wave (CDW) [76]. It was proposed that the CDW originates from the flat phonon modes with Sn vibrations [77]. Interestingly, ScV_6Sn_6 also exhibits the spontaneous TRSB state [78]. The mechanism of the TRSB state in ScV_6Sn_6 is an interesting future problem. (Note that the contribution from the vHS points is not a requirement for the cLC order [67].) The GL free energy analysis was performed in Ref. [79]. Recently, very weak but definite signal of the nematic electronic order has been observed in Ti-based kagome metal CsTi_3Bi_5 [80, 81]. To explain the observed hidden nematicity, the “odd-parity BO” without TRSB has been predicted theoretically [82].

Important technical details:

We have added more detailed explanations in the revised SI for the readers who have an interest in the important technical details, by following the useful advice from Reviewer #1. However, the additions to the main text were kept to a very small amount for the sake of readability for a more general readership.

Important recent experiments closely related to the present theory:

Very recently, strain-free highly symmetric fabricated CsV_3Sb_5 samples were studied in arXiv:2304.00972; Ref. [74]: It was proposed that CsV_3Sb_5 is located at the quantum critical point of the current order ($T_{\text{TRSB}} \approx 0$) in the absence of the uniaxial strain. The field-induced (hz~9 Tesla) current order at $T_{\text{TRSB}} \sim 20$ K is naturally understood based on the GL free energy analysis with the current-bond- h_z trilinear term [75]. Also, the present GL theory explains the field-induced

enhancement of the local magnetic flux ($\propto \eta^1$) observed by μ SR measurements in AV_3Sb_5 [15, 17, 18].

Thus, with the emergence of these new experimental facts, the present theory is becoming increasingly important. In the revised manuscript, we added the following paragraph about key experimental reports on page 11:

“Here, we shortly discuss several experimental evidences of the BO+cLC coexistence. Recent transport measurement of highly symmetric fabricated CsV_3Sb_5 micro sample [74] reveals that small magnetic field h_z (<10 T) or small strain gives rise to the nematic BO+cLC coexisting state below T_{BO} . This finding is well explained by the recent GL theory under h_z [75]: The current-bond- h_z trilinear coupling caused by the orbital magnetization gives rise to the sizable h_z -induced cLC order in the BO state. This theory also explains h_z -induced enhancement of the cLC order observed by μ SR measurements [15,17,18] and field-tuned chiral transport study [20].”

○ Comment #1-2

Below I provide responses to the authors answers in the cases where I believe there are still outstanding issues:

1–3: If I understand correctly, the authors argue that bond order is dominant in the 3–site kagome model near the upper van Hove singularity, and that this therefore provides a good starting point for the study of how bond–order fluctuations develop in the AV_3Sb_5 kagome metals. I have several problems with this argument. First, in the stated references [6,7,11], I am not sure exactly which parts the authors refer to. In [6,7], both fRG studies, the phenomenology is richer than only bond order, and there are multiple instabilities that the authors do not consider in the present manuscript. This fact should be made apparent in the text, bond order is not the only instability occurring near the pure van Hove singularity, it is very sensitive to interactions, as also shown in, e.g., Ref 10. In [11], if I understand correctly, the authors find that the naïve 3–site model does actually not support bond order unless additional interactions are included (coming from specific Maki–Thompson and Azlamazov–Larkin terms). From Ref. [11] it is not clear to me why one should expect such terms to play an important role in either the AV_3Sb_5 compounds or in the 3–site kagome model. Furthermore, while I appreciate that the authors aim to study the impact of bond–order fluctuations, they are not ruling out that other effects could be responsible for the experimental observations. Indeed, as they start from Eq. (2), they only allow bond–order fluctuations to contribute and neglect the influence from any other channel. Allowing inter–

and intra-sublattice channels to compete by including an additional Fermi surface would provide a more complete picture of the AV3Sb5 materials.

○ Reply #1-2

We are grateful to Reviewer #1 for the fruitful comment on the rich quantum phases in kagome metals. We consider that various exotic quantum orders can be realized in kagome metals because conventional antiferromagnetic order is suppressed by the geometrical frustration, which is closely related to the “sublattice interference” in the band structure. Below, we would like to reply to the comments in more detail.

BO instability obtained by fRG theories:

Here, we explain the functional RG study in Refs. [6,7]. Fig. R1 in this Reply shows the phase diagram of kagome lattice extended Hubbard model with onsite U (or U_0) and nearest-neighbor-site V (or U_1) Coulomb interaction. cBO = BO in the present manuscript. In both (left) and (right), the BO phase is realized for $U \gg 1$ ($=2|t|$) and $V/U \ll 1$. We stress that the BO phase is robustly realized even for “ $V=0$ ” in (right), which is consistent with our theory of the AL processes for “ $U \gg 1$ and $V=0$ ” in Ref. [12]. Thus, the experimental BO phase is robustly reproduced based on the realistic interaction Hamiltonian ($U \gg 1$ and $V/U \ll 1$) in both the functional RG method [6,7] and our previous study [12].

In the parquet RG study in Ref. [10], the authors performed the analysis for “more general interaction Hamiltonian (composed of $g_1^0, g_2^0, g_3^0, g_4^0$)” and demonstrated that rich exotic states can emerge depending on the initial interactions. This is a very significant theoretical analysis. In the present study, however, we adopt the “BO interaction Hamiltonian with single parameter v ” in Eq. (2). Its reliability and the origin v is explained on pages 4-5 in the main text. We stress that experimentally observed BO is actually obtained by the parquet RG theory for the present interaction Hamiltonian; see Fig. S11 in the SI F.

To summarize, in the present manuscript, we concentrate on the present BO interaction Hamiltonian to focus on the “analysis of the current+bond order mechanism”.

Why do AL processes give large BO instability ?:

We would like to explain the significance of the Aslamazov-Larkin (AL) processes revealed in Ref. [12]. The AL processes naturally explain the electronic nematic states in Fe-based SCs [36] and twisted-bilayer-graphene [44]. Its various applications and the theoretical basis are explained in our recent review article [45]. The AL processes are directly derived from the Baym-Kadanoff

conserving scheme. We will explain its importance based on the numerical analysis in Reply #1-3.

We stress that we have studied a realistic 30 orbital (15 d-orbital + 15 p-orbital) model for AV_3Sb_5 in Ref. [12]. Based on the density-wave equation analysis, we obtained the BO instability at q_1 , q_2 and q_3 , as shown in Fig. 6 c in Ref. [12]. Thus, we consider that the AL processes are the origin of the BO in real AV_3Sb_5 ,

Impact of inter-sublattice channels:

As pointed out the Reviewer #1, there are several significant theoretical papers on the current order mechanism. Especially, some papers propose interesting effects of the electron correlation of non- b_{3g} d-orbitals and the inter-orbital order parameters. In the revised manuscript, we added the following introduction of other theories:

“In the present study, we focus on the pure-type band composed of b_{3g} -orbital. However, the impact of other 3d-orbitals on the cLC order has also been studied in Refs. [30,73]. The extension of the present theory to multi-orbital models is a very important future issue.”

Other theoretical studies on cLC mechanism:

We have added the following important theoretical papers in the revised manuscript with appropriate explanations:

[11] Y.-P. Lin and R. M. Nandkishore, Phys. Rev. B 104, 045122 (2021).

[30] M. H. Christensen, et al., Phys. Rev. B 106, 144504 (2022).

[73] H. D. Scammell, et al., Nat. Commun. 14, 605 (2023).

[77] H. Hu, et al, arXiv:2305.15469.

[79] F. Grandi, et al., Phys. Rev. B 107, 155131 (2023)

○ Comment #1-3

1-4: I understand the interaction in Eq. (2) ensures that only bond-order contributions enter the susceptibility since v only parameterizes this interaction. However, it is very hard for the reader to know what v corresponds to microscopically (i.e. in terms of e.g. Coulomb interactions). I think the paper would improve significantly if the authors would provide a relationship between v and electronic interactions in the main text. For instance, the interaction between the bond-order fluctuations in the patch model of Ref. 10 are ascribed to specific intra- and inter-patch interactions, which makes it very easy for the reader to identify the ingredients necessary to promote such bond-order fluctuations. Would it be

possible for the authors to provide a similar relationship between interactions and their parameter ν ? They hint at it below Eq. (2) but it is never made explicit. The associated supplementary B-2 is very opaque and hard to follow. In my opinion, the authors could improve the text by integrating part of that supplementary into the main text and make succinct statements about how ν is related to the microscopic parameters both in the exact and approximate scenarios. This would strengthen the paper and make it easier for the reader to understand. While I appreciate y is a bit harder to pin down, I think it would make the results more compelling if the authors explored a larger parameter regime in y to demonstrate the robustness of their findings. For instance, by making a three-dimensional plot containing ν , y , and T (Figs. 4 (d) and (e) provides an indication but more values of y is desirable).

○ Reply #1-3

We are grateful to Reviewer #1 for these very important questions about the values of ν and y . As we discussed in Reply #1-2, the AL processes cause the BO phase in kagome metals. Here, we explain the BO interaction used in the present manuscript, $\nu=1 \sim 1.5$, is actually given by the AL processes by referring to Ref. [12]. We also explain that the relation $y \geq 0.5$ is realistic in real systems.

Effective interaction given by AL processes:

Figure R2 shows the (a) AL1 and (b) AL2 processes. The former (latter) gives large g_{back} (g_{um}). Near the BO critical point, the eigenvalue of the BO instability λ_{bond} is slightly smaller than unity. In this case, the obtained $[g_{\text{back}}+g_{\text{um}}]/2$ is about 1.5, which is consistent with the value of ν used in the present manuscript.

To add the abovementioned discussion in the revised manuscript on page 5 as:

“In (i), Eq. (2) is induced by the spin-fluctuation-mediated beyond-RPA processes, the importance of which was originally revealed in the study of Fe-based SCs [45]. Its diagrammatic expression and numerically obtained BO fluctuation-mediated interaction are shown in Figs. 3 (a)-(e) in Ref. [12]. The parameter ν in Eq. (2) is given as $[g_{\text{back}} + g_{\text{um}}]/2$, which is about 1.5 near the BO critical point ($\lambda_{\text{bond}} \lesssim 1$). Thus, the value of ν given by the AL processes is comparable to that used in the present study.”

Why $y \geq 0.5$ is realistic in real kagome metals ?

In the previous manuscript, we explained that the parameter y in the MT-type interaction in Eq. (S18) depends on the BO fluctuation mechanism: We obtain $y=2$ for the off-site Coulomb interaction in Eq. (S4) because both charge- and spin-channel BO fluctuations develops as we

explain in the SI B-2. To improve the explanation more plainly, we have added the detailed discussion for the first-order and second-order diagrams step-by-step with the aid of new figures in Fig. S2 c.

On the other hand, we obtain $y=0.5$ for the BO interaction in Eq. (2) because only charge-channel BO fluctuations develop. For the same reason, $y\sim 0.5$ for the AL process mechanism (because the spin BO instability is very small [12]), and this mechanism gives the effective coupling constant $v_{AL} \sim [g_{back}+g_{um}]/2 \sim 1.5$. (A misleading explanation in the previous manuscript was corrected.) We added this explanation in the SI “B-3: The relation $y \geq 1/2$ when v and V coexist”.

In the revised SI, we discuss the value of y when the BO interaction v and the off-site Coulomb V coexist, in the SI B-3. Here, we explain the outline: The spin- and charge-BO Stoner factors, α^s and α^c , are equal when $V>0$ and $v=0$. In this case, we obtain $y=2$. With introducing $v>0$, however, α^c increases while α^s is unchanged. Therefore, $I^c \propto 1/(1-\alpha^c)$ becomes much larger than $I^s \propto 1/(1-\alpha^s)$ near the charge-BO critical point $\alpha^c \sim 1$. For this reason, the total MT term is $I^{MT} = [I^c + 3 I^s]/2 \geq I^c/2$, and therefore $y \geq 0.5$.

In Fig. R3, we show the numerical result for $y=1.5$, where the charge-current order is robustly obtained. However, we consider that $y>1$ is unrealistic, so we did not include this result in the revised manuscript.

○ Comment #1-4

1-6: I think my confusion in this question stemmed from the authors referring to the nematic state as a 3Q state which, in my mind implies that three order parameters have the same magnitude. I appreciate now that this is not what the authors had in mind. Regarding the schematic phase diagram, only the leading instability is obtained from the density-wave equation, the number of Q-vectors condensing is, as far as I can tell, determined from the Landau free energy. Since the number of possible phases is vast, the authors should provide the details of the calculations of the coefficients of the free energy which supports their assertions that the stabilized phases are indeed the ones appearing in the schematic phase diagram.

○ Reply #1-4

As mentioned by Reviewer #1, various phases can appear in kagome metal depending on the initial interaction parameters (e.g., g_1-g_4 in parquet RG). In the present manuscript, however, we focus on the study of the “mechanism of the current+bond order states”. “The uniform coexistence

of the current and bond orders” can be realized when the relation $R \cdot R' > 1$ is satisfied, where $R = d_{1,a}d_{2,a}/(d_{3,a})^2$ and $R' = d_{1,b}d_{2,b}/(d_{3,b})^2$. This important relation is clearly satisfied in Fig. S9 **d.** in the present numerical study.

In the SI E-2, we obtained the phase diagram in Fig. S9 a, by minimizing the GL free energy of 13^2 kinds of order parameters, $(\phi, \eta) = (\phi_m, \eta_n)$ with $m, n = 1 \sim 13$. The bond+current coexisting state has C_6 symmetry only when $(\phi, \eta) = (\phi_1, \eta_1), (\phi_2, \eta_2), (\phi_3, \eta_3)$, while the other 166 coexisting states have lower (C_2) symmetry. Under this 13^2 basis, the C_6 symmetry BO+cLC phase is already eliminated. Importantly, by considering an additional 14th BO parameter $\phi_{14} \propto (\phi, \phi, \phi')$, the area of the C_2 symmetry BO+cLC phase is widened, as we show in Fig. S9. Thus, the area of the “ C_2 symmetry BO+cLC phase” in Fig. S9 is still “underestimated” in the present calculation. Therefore, this result reinforces the existence of the “ C_2 symmetry BO+cLC phase” in Fig. S9 that is one of the main findings of the present theory. It is an important future problem to minimize the GL free energy more completely to know the accurate area of the C_2 symmetry BO+cLC phase.

We believe that the method of calculating the GL coefficients and the details of the numerical calculation has already been explained in SI E-1 and E-2 in sufficient detail. Because the main subject of this paper is the “microscopic mechanism” of the current order, we would like to leave the further detailed analysis of the GL coefficients for future analysis.

○ Comment #1-5

1–13: This is the site-symmetry group of the vanadium atoms (or more generally, the sites of the kagome lattice). $D6h$ is the point group.

○ Reply #1-5:

We are grateful to Reviewer #1 for his/her previous comment. We have already corrected the statement in the previous manuscript: “The d-orbital belongs to b_{3g} of the D_{2h} point group at V site.”

Reply to Comments by Reviewer #2

○ Comment #2-1

The revised manuscript by Rina Tazai et al have improved significantly to make their proposal of BO fluctuation induced cLC order more convincing. All my detailed questions and comments have been addressed carefully by the authors. Therefore, I would recommend the manuscript to be publish in Nature Communications.

○ Reply #2-1

We are really grateful to Reviewer #2 for his/her very constructive and enlightening comments, which were really useful in revising the manuscript. We performed the RG analysis in the SI F thanks to the suggestive comments by Reviewer #2. Thank you very much for evaluating the 2nd manuscript and for recommending the manuscript for publication.

○ Comment #2-2

While reading the revised manuscript, I noticed some of the notations remain undefined. While the readability of the manuscript has been improved, I believe the authors should make another effort to check the presentation of the work and make sure all quantities have been defined properly in the text (say a parameter y in figure 4 (d, e)), and all the figures referred in the text are presented (say Fig. S2 d referred in the SM is not shown). Such effort would increase the impact of the work.

○ Reply #2-2

Thank you very much for these kind comments. We fixed the problems pointed out by Reviewer #2. We have also made further efforts to improve the manuscript. We believe that the latest manuscript becomes much more readable and therefore appropriate for publication.

Reply to Comments by Reviewer #3

○ Comment #3-1

The authors have discussed the criticism by reviewers thoroughly and very diligently. The revision improves the quality of the paper. I recommend this manuscript to be published in Nature Communication as it is.

○ Reply #3-1

We are really grateful to Reviewer #3 for his/her very useful comment about the relation between the present theory and several key experiments. In the 2nd manuscript, we have improved the numerical analysis to explain the experimental phase diagram thanks to the suggestive comments by Reviewer #3. Thank you very much for evaluating the last submitted manuscript and for recommending the manuscript for publication.

Fig. R1: Phase diagram of kagome lattice extended Hubbard model with onsite (U_0 or U) and nearest-neighbor-site (U_1 or V) Coulomb interaction. cBO represents the BO state in the present manuscript. (left) from Ref. [6]. (right) from Ref. [7].

Fig. R2: (a) AL1 and (b) AL2 processes. The former (latter) gives large g_{back} (g_{um}). (c) λ_{bond} , g_{back} and g_{um} as functions of α_S . $\alpha_S=0.75$ corresponds to $U=1.26$. From Ref. [12].

Fig. R3:

Obtained T_{cLC} and T_{BO} as functions of v for $y=1.5$.

The current order is robustly obtained.

However, we consider that $y = 0.5 \sim 1.0$ is realistic in kagome metals, as we discuss in the revised manuscript. For this reason, this figure is not include in the revised manuscript.

List of main changes made in the revised manuscript:

The main changes made in the revised manuscript have been explained in the Reply to each Reviewer. The revised parts are shown in red characters in the revised manuscript.

--- **Main text** ---

○ **page 5**

Explanation that the BO interaction v in Eq. (1) is given by the AL processes, by referring to Figs. 3 (a)-(e) in Ref. [12].

○ **page 7**

Explanation that the parameter y becomes ~ 0.5 in real kagome metals, by referring to the revised SI B-2.

○ **page 11**

Discussion about the relation between the present theory and the RG theory.

○ **page 11**

Summary section:

- Introduction of other theories [30,73] that focus on the impact of other 3d-orbitals on the cLC order.
- Discussions about experimental evidences of the BO+cLC coexisting states.
- Introduction of other kagome metals: cLC order in ScV6Sn6, nematic order in CsTi3Bi5.

○ **Added references**

[11] Y.-P. Lin and R. M. Nandkishore, Phys. Rev. B 104, 045122 (2021).

[30] M. H. Christensen, et al., Phys. Rev. B 106, 144504 (2022).

[73] H. D. Scammell, et al., Nat. Commun. 14, 605 (2023).

[77] H. Hu, et al, arXiv:2305.15469.

[79] F. Grandi, et al., Phys. Rev. B 107, 155131 (2023)

---**Supplementary Information** ---

○ We revised “**B-2: Effective interaction due to BO susceptibility**” very eagerly to make the explanation much more readable, by introducing revised **Fig. 2 (c)**.

○ We added a new section “**B-3: The relation $y \gtrsim 1/2$ when v and V coexist**” to explain that the coexistence of V (= off-site Coulomb) and v (= BO interaction) leads to the relation $y \sim 0.5$

REVIEWERS' COMMENTS

Reviewer #1 (Remarks to the Author):

I have read the authors' replies to my questions. While I appreciate the findings presented in the manuscript, I still do not find that they are of sufficient merit and generality to support publication in Nature Communications over a more specialized journal. While the authors state that their idea and results will be useful in various metals, this claim is not supported by any of the material presented subsequently, and it is not clear how the results aid in the understanding of current orders in materials beyond the AV3Sb5 kagome metals. Finally, I still find the readability of the manuscript a serious issue, as also remarked upon by another referee. It has certainly improved over the first version, but it remains my impression that a physics educated reader of Nature Communications will have a very hard time following the steps presented.

The authors addressed most of previous comments. One issue that still remains also relates to the generality of the results presented. The authors compare their results favourably to those of Refs. 6 and 7 and using this agreement as support for considering exclusively bond order fluctuations. However, I found a much less convincing agreement between the studies. The phase diagrams reproduced in the supplementary from Refs. 6 and 7 are for chemical potentials corresponding to hexagonal Fermi surfaces which exhibit perfect nesting. The present manuscript studies a Fermi surface which does not exhibit perfect nesting since a next-nearest neighbour hopping parameter is included. Hence, the studies cannot be directly compared. Moreover, as shown in Ref. 6, the bond order (or CBO) phase is very sensitive to the geometry of the Fermi surface and is completely absent from the phase diagram when the Fermi surface is changed just a bit from a hexagonal shape (Fig. 2(b) of Ref. 6). Ref. 7 did not study the phase diagram when the Fermi surface is not hexagonal. This is at odds with the authors' claimed agreement between the different studies. The authors' own parquet RG study starts from a patch model and thus assumes that the van Hove points are the main drivers of the phenomenology but based on the findings of, e.g., Ref. 6, it is unclear if this approach is valid if the Fermi surface is not hexagonal and perfectly nested. While this appears a minor detail, it raises the question of how general the results presented are and is thus relevant in the context of deciding whether the manuscript merits publication in Nature Communications.

Reply to Comments by Reviewer #1

○ Comment 1-1:

I have read the authors' replies to my questions. While I appreciate the findings presented in the manuscript, I still do not find that they are of sufficient merit and generality to support publication in Nature Communications over a more specialized journal. While the authors state that their idea and results will be useful in various metals, this claim is not supported by any of the material presented subsequently, and it is not clear how the results aid in the understanding of current orders in materials beyond the AV3Sb5 kagome metals. Finally, I still find the readability of the manuscript a serious issue, as also remarked upon by another referee. It has certainly improved over the first version, but it remains my impression that a physics educated reader of Nature Communications will have a very hard time following the steps presented.

○ Reply 1-1:

We are grateful to the statement in Comment 1-1 “*I appreciate the findings presented in the manuscript.*” The previous manuscript has been highly evaluated and recommended for publication in Nature Communications by Reviewers #2 and #3. To support the “*sufficient merit and generality to support publication*” of this study, here we introduce very recent notable experimental reports that appear after the previous resubmission, which are satisfactorily understood by the present theory:

(1) Emergence of “**single-Q current order state at $T^*=130\text{K}$** ” reported by T. Asaba et al [21]. (submitted to arXiv on September 29, accepted for publication in Nature Physics)

(2) Observation of “**out-of-phase combination of bond charge order and loop currents**” reported by Y. Xing et al. [68] (submitted to arXiv on August 8)

As for (1), the observation at T^* is naturally explained based on Fig. 4 f by considering that the system is in the region $1/v > 1/v^*$. In the revised manuscript, we added the following explanation on page 10 in the main text: “**In contrast, the 1Q cLC state is realized when $2d_{2,a}/d_{2,b}$ is smaller than unity. Thus, the electronic state becomes nematic at $T_{\text{cLC}} (>T_{\text{BO}})$. In this case, there is no secondary BO component above T_{BO} . Recently, strong evidence of the emergence of the 1Q cLC state at $\sim 130\text{K} (> T_{\text{BO}})$ has been reported by the magnetic torque measurement [21].**”

As for (2), the observed ordered state in Ref. [68] is essentially the same as our theoretical prediction: “Nematic BO+cLC state” in Fig. 5 c. In the revised manuscript, we added the

following explanation on page 10 in the main text: “This result is consistent with the recent observation of out-of-phase combination of bond charge order and loop currents by STM measurement [68].” We also added the following explanation on page 12 in the main text: “It is noteworthy that the nematic electronic state that supports the C_2 BO+cLC order in Fig. 5 c has been reported by recent STM measurement [68].”

By these revisions, the latest revised manuscript possesses the “*sufficient merit and generality to support publication in Nature Communications*”.

○ Comment 1-2:

The authors addressed most of previous comments. One issue that still remains also relates to the generality of the results presented. The authors compare their results favourably to those of Refs. 6 and 7 and using this agreement as support for considering exclusively bond order fluctuations. However, I found a much less convincing agreement between the studies. The phase diagrams reproduced in the supplementary from Refs. 6 and 7 are for chemical potentials corresponding to hexagonal Fermi surfaces which exhibit perfect nesting. The present manuscript studies a Fermi surface which does not exhibit perfect nesting since a next-nearest neighbour hopping parameter is included. Hence, the studies cannot be directly compared. Moreover, as shown in Ref. 6, the bond order (or CBO) phase is very sensitive to the geometry of the Fermi surface and is completely absent from the phase diagram when the Fermi surface is changed just a bit from a hexagonal shape (Fig. 2(b) of Ref. 6). Ref. 7 did not study the phase diagram when the Fermi surface is not hexagonal. This is at odds with the authors’ claimed agreement between the different studies. The authors’ own parquet RG study starts from a patch model and thus assumes that the van Hove points are the main drivers of the phenomenology but based on the findings of, e.g., Ref. 6, it is unclear if this approach is valid if the Fermi surface is not hexagonal and perfectly nested. While this appears a minor detail, it raises the question of how general the results presented are and is thus relevant in the context of deciding whether the manuscript merits publication in Nature Communications.

○ Reply 1-2:

The main concern of Reviewer #1 in this paragraph is about “**the robustness of the BO solution when the system deviates from the perfect nesting**”. We understand the importance of this comment. It is noteworthy that this concern was already answered in our previous research based on the DW equation analysis [12]: In Fig.6 of Ref. [12], we robustly obtained the BO solution for the “first principles 30 orbital model for CsV_3Sb_5 , which is far from the perfect

nesting due to long-range hopping integrals. To confirm the robustness of the BO solution more completely, we added the discussions about the renormalization-group theory:

1. As suggested by Reviewer #1, the BO solution seems to be more fragile in the fRG study in Ref. [6] for n deviates from the van Hove filling n_{VHS} . (We consider that there is insufficient information to assert that the perfect nesting is a necessary condition for the BO solution in Ref. [6].) However, the BO instability in Ref. [6] (= fRG study) is “**underestimated**” because Ref. [6] assumes that (A) the wavevector of the BO is fixed to be commensurate, and (B) the BO form factor has a simple nearest-neighbor component. These assumptions are invalid when the nesting becomes worse for $n \neq n_{\text{VHS}}$. In contrast, **both the BO wavevector and BO form factor are fully optimized to maximize the transition temperature** in the DW equation method in Ref. [12]. Therefore, the present DW equation method [12] that is free from assumptions (A) and (B) gives reliable results even for $n \neq n_{\text{VHS}}$, while Ref. [6] would give a serious underestimation of the BO instability for $n \neq n_{\text{VHS}}$.

Importantly, the “improved fRG theory in Ref. [66]” developed by the present authors enables us to optimize the wavevector and the form factor. Therefore, we can verify the robustness of the BO solution by using the improved fRG method [66]. This is our important future study plan.

On page 5 in the revised main text, we added the following sentences: “A great advantage of this theory [12] is that both the functional form of the BO form factor and the BO wavevector are automatically optimized to maximize T_{BO} . Based on this theory, the BO at $\mathbf{q}=\mathbf{q}_n$ ($n=1,2,3$) is robustly obtained based on the first principles multiorbital model for CsV_3Sb_5 [12].”

2. We also explain the robustness of the BO solution based on the parquet RG method in the revised Supplementary Note 5. Here, the parameter d ($0 < d < 1/2$) in the parquet RG represents the strength of the nesting. We set $d=1/2$ in the previous Supplementary Note 5. To verify the robustness of the BO solution, here we set **$d=1/4$** in the revised Supplementary Note 5: **Even in this case ($d=1/4$), the BO solution is robustly obtained even** as shown in the revised Supplementary Fig. 6.

Therefore, we conclude that the perfect nesting of the Fermi surface is not a necessary condition for the BO solution. To explain this, we added the following sentence in the revised Supplementary Note 5: “Note that E corresponds to $\sim T$, so $T \approx E_0 \exp(-z^{1/2})$. When $E_0 = 0.2\text{eV}$, $z = 10$ corresponds to $T \approx 100\text{K}$. The parameter d is bounded $0 < d < 1/2$, and $d = 1/2$ corresponds to the perfect nesting. Hereafter, we set $d = 1/4$ because the nesting of the real FS is not perfect. In this study, strong cLC and BO instabilities are robustly obtained for $d = 1/4 \sim 1/2$.”

By these revisions, the latest revised manuscript possesses the “*sufficient merit and generality to support publication in Nature Communications*”.

List of main changes made in the revised manuscript:

The main changes made in the revised manuscript have been explained in the Reply to Reviewer #1. The revised parts are shown in red characters in the revised manuscript. Other important changes made in the revised manuscript are as follows:

○Figure 4 f: We have replaced the words “3Q cLC+BO” with “cLC” because the 1Q cLC state can be realized in our theory. The present theory presents a natural explanation for the **1Q cLC order** observed by magnetic torque measurement in Asaba et al [21]. We added the following sentence in the revised main text on page 10: “**In contrast, the 1Q cLC state is realized when $2d_{2,a}/d_{2,b}$ is smaller than unity. Thus, the electronic state becomes nematic at $T_{cLC} (>T_{BO})$. In this case, there is no secondary BO component above T_{BO} . Recently, strong evidence of the emergence of the 1Q cLC state at $\sim 130K (>T_{BO})$ has been reported by the magnetic torque measurement [21]**” In addition, we also replaced the words “3Q BO” and “3Q BO+cLC” with “3Q-BO (in-plane C_6)” and “BO+cLC (in-plane C_2)” in Figure 4 f, respectively, to improve the quality of this schematic phase diagram.